# Elucidating electrochemical nitrate and nitrite reduction over atomically-dispersed transition metal sites

Eamonn Murphy[1,7], Yuanchao Liu [1,7], Ivana Matanovic[2,3], Martina Rüscher[4], Ying Huang [5], Alvin Ly[5], Shengyuan Guo [1], Wenjie Zang [5], Xingxu Yan [5], Andrea Martini[4], Janis Timoshenko [4], Beatriz Roldán Cuenya [4], Iryna V. Zenyuk [1,5], Xiaoqing Pan [5], Erik D. Spoerke [6] & Plamen Atanassov [1,5] ✉

Electrocatalytic reduction of waste nitrates ($NO_3^-$) enables the synthesis of ammonia ($NH_3$) in a carbon neutral and decentralized manner. Atomically dispersed metal-nitrogen-carbon (M-N-C) catalysts demonstrate a high catalytic activity and uniquely favor mono-nitrogen products. However, the reaction fundamentals remain largely underexplored. Herein, we report a set of 14; 3$d$-, 4$d$-, 5$d$- and $f$-block M-N-C catalysts. The selectivity and activity of $NO_3^-$ reduction to $NH_3$ in neutral media, with a specific focus on deciphering the role of the $NO_2^-$ intermediate in the reaction cascade, reveals strong correlations (R=0.9) between the $NO_2^-$ reduction activity and $NO_3^-$ reduction selectivity for $NH_3$. Moreover, theoretical computations reveal the associative/dissociative adsorption pathways for $NO_2^-$ evolution, over the normal M-$N_4$ sites and their oxo-form (O-M-$N_4$) for oxyphilic metals. This work provides a platform for designing multi-element $NO_3$RR cascades with single-atom sites or their hybridization with extended catalytic surfaces.

Recently, significant effort has been focused to decarbonize the production of chemicals and fuels[1,2]. Among these, the decarbonization of ammonia ($NH_3$) synthesis from the traditionally energy intensive and environmentally damaging Haber-Bosch (H-B) process remains a grand challenge[3,4]. The electrochemical synthesis of $NH_3$, can ideally utilize renewable energy, reactive N-species and protons from water to generate $NH_3$. The reduction of di-nitrogen ($N_2$) in aqueous protic electrolytes remains the most heavily researched green pathway for $NH_3$ synthesis[5]. However, due to the high bond dissociation energy of $N_2$ (945 kJ mol$^{-1}$), ultra-low solubility in aqueous electrolytes and competition from the hydrogen evolution reaction (HER), the current Faradaic efficiencies (FE) and $NH_3$ yield rates (Yield$_{NH3}$) remain

prohibitively low[4,6]. Given the total lack of reproducibility and the difficulties of ubiquitous $NH_3$ contamination in electrochemical systems, direct $N_2$ reduction remains unproven in aqueous protic electrolytes[7].

To circumvent the challenges of $N_2$ reduction, there is a renewed interest in closing the N-cycle by recycling the reactive, more oxidized N-species (nitrate-$NO_3$, nitrite-$NO_2$ and nitric oxide-NO) to $NH_3$[8–11]. The electrochemical $NO_3^-$ reduction reaction ($NO_3$RR) is of particular interest due to its weaker N=O bond (204 kJ mol$^{-1}$), large solubility in aqueous electrolytes and widespread availability. $NO_3^-$ is an environmental pollutant found in industrial waste streams and agricultural runoffs, due to the heavy overfertilization practices currently utilized[8].

---

[1]Department of Chemical and Biomolecular Engineering, University of California, Irvine, CA 92697, USA. [2]Department of Chemical and Biological Engineering, University of New Mexico, Albuquerque, NM 87131, USA. [3]Theoretical Division, Los Alamos National Laboratory, Los Alamos, NM 87545, USA. [4]Department of Interface Science, Fritz Haber Institute of the Max Planck Society, 4-6 Faradayweg, Berlin 14195, Germany. [5]Department of Materials Science and Engineering, University of California, Irvine, CA 92697, USA. [6]Sandia National Laboratories, Albuquerque, NM 87185, USA. [7]These authors contributed equally: Eamonn Murphy, Yuanchao Liu. ✉e-mail: plamen.atanassov@uci.edu

While the NO₃RR to NH₃ provides a pathway for efficient recycling of NH₃/NO₃⁻ (originating from the H-B process), additional efforts are focusing on low energy plasma techniques to produce NO₃⁻ from air, but such processes are to date, not energy efficient[12–14].

The electrochemical NO₃RR to NH₃ is a complex $8e^-$ ($9H^+$) transfer process, involving several desorbable and non-desorbable surface intermediates, creating a challenge for tailoring the catalytic surface for favorable adsorption of specific intermediates due to scaling relations[15–17]. Previous attempts were made to tune the intermediate adsorption binding energies by modulating the electronic structure on extended metal surfaces through alloys such as CuNi, PdAu, and PtRu[18–20]. These attempts achieved improved FE_NH₃ and Yield_NH₃ compared to mono-metallic counterparts, however, these approaches are still constrained by scaling relations and combat this by utilizing strongly alkaline or acidic environments with large concentrations of NO₃⁻.

In the biological NO₃RR, the reaction is partitioned as an enzymatic cascade, where the initial $2e^-$ reduction of NO₃⁻-to-NO₂⁻ is catalyzed over a Mo-cofactor in nitrate-reductase and the further reduction of NO₂⁻ to NH₃/N₂ is catalyzed over an Fe- or Cu-cofactor in nitrite-reductase[21–23]. Inspired by the enzymatic pathway, recent work demonstrated that an atomically dispersed bi-metallic FeMo-N-C catalyst could successfully partition the NO₃RR into the NO₃⁻-to-NO₂⁻ and NO₂⁻-to-NH₃ constituents, outperforming its mono-metallic Mo-N-C or Fe-N-C counterparts. The atomically dispersed Mo-N_x sites facilitated dissociative adsorption of NO₃⁻, releasing NO₂⁻, which is subsequently re-adsorbed and reduced to NH₃ over Fe-N_x sites at 100% FE[24]. Additionally, other approaches to partition the NO₃RR are emerging in the field for example, employing heterogenous metal sites, where each metal is tailored to efficiently catalyze a segment of the reaction. A recent study demonstrated potential dependent phase transitions of a Cu-Co (3–5 nm) binary metal sulfide catalyst, in which the inner Cu/CuO_x phases efficiently reduce NO₃⁻-to-NO₂⁻, while the outer Co/CoO phases selectively reduce the NO₂⁻-to-NH₃. The latter results in a NO₃RR cascade to NH₃, reaching a high FE of 90.6%[25]. A recent complementary work examining a series extended of transition metal surfaces by Carvalho et al. described NO₃RR catalyst design parameters by linking the electronic structure to experimentally observed NO₃RR performance[26]. These design parameters are derived from microkinetic models that described the $2e^-$ transfer rate limiting step (NO₃⁻-to-NO₂⁻) and further the NO₃RR FE_NH₃, identifying a competitive adsorption between H⁺ and NO₃⁻, that was described by the material-dependent property ($\Delta G_{H^+} - \Delta G_{NO_3^-}$). The study found that increasing FE_NH₃ correlates with NO* binding and subsequent dissociation into N* and O* as the d-band energy approaches the Fermi energy, resulting in the Co-foil showing the highest ammonium selectivity.

Atomically dispersed metal-nitrogen-carbon (M-N-C) catalysts have been extensively studied in neighboring electrocatalytic reactions (carbon dioxide reduction-CO₂RR[27–29], oxygen reduction-ORR[30,31]), demonstrating excellent catalytic activities and unique reaction pathways. Recently, atomically dispersed Fe-N-C catalysts were shown to be very selective for the NO₃RR to NH₃[32,33], owing to their maximized atomic utilization and favorable NO₃⁻ adsorption over H⁺. Additionally, isolated active sites provide an unfavorable environment for coupling adsorbed N-molecules, enabling mono N-based products (e.g., NH₃) to be selectively produced.

The current understanding of atomically dispersed M-N-C catalysts for the complex NO₃RR is very limited, with only M = Fe, Mo, and Cu being explored at varying conditions. Herein, we identify key transition metals and rare earth elements, important for electrocatalytic reactions and synthesize a set of 13 atomically dispersed M-N-C catalysts (M = Cr, Mn, Fe, Co, Ni, Cu, Mo, Ru, Rh, Pd, W, La, and Ce). The reaction onset potential (HER, NO₃RR and NO₂RR), the NO₃RR selectivity to both NO₂⁻ and NH₃ and the NO₂RR selectivity to NH₃, elucidate several experimental activity descriptors. Isotopically doped

¹⁵NO₂⁻ in the NO₃RR highlights the complex production/consumption mechanism of the NO₂⁻ intermediate. Density functional theory (DFT) evaluates the Gibbs free energy for both the NO₃RR and NO₂RR following either the dissociative adsorption or associative adsorption pathway. Relating the experimentally determined and computationally determined activity descriptors, reveals strong correlations for the NO₂RR ($R = 0.72$) and the NO₃RR ($R = 0.73$) selectivity to NH₃. This work bridges the gap between computation and experiment for the NO₃⁻ and NO₂⁻ reduction reactions over atomically dispersed M-N-C catalysts by providing a powerful set of activity descriptors that correlate strongly with the experimentally observed activity. These descriptors can be utilized to guide future atomically dispersed M-N_x catalyst development, for highly active and efficient NO₃⁻ reduction to NH₃.

## Results

### Structural environment of single-atom metal centers

Atomically dispersed M-N-C catalysts were synthesized through the well-established sacrificial support method (SSM) originally developed by our group[34–36]. The SSM has been extensively applied for atomically dispersed Fe-N-C catalysts for the ORR and CO₂RR[27,37]. Here, extending beyond Fe-N-C to a variety of 3d-, 4d-, 5d- and f-block metals, a set of atomically dispersed M-N-C catalysts (M = Cr, Mn, Fe, Co, Ni, Cu, Mo, Ru, Rh, Pd, La, Ce, and W) were synthesized, as shown in Fig. 1a. The metallic centers ideally have a first coordination shell of 4-nitrogen atoms and are coordinated in either an in-plane or out-of-plane configuration as shown in Fig. 1b. With the synthesis conditions being adjusted to maintain the atomically dispersed nature for each metal element (Table S1), the resulting catalysts showed consistent porosity and pore size distribution (Fig. S1), degree of graphitization (Fig. S2) and a well-maintained metal loading at 0.5–1.5 wt% (Fig. S3).

X-ray diffraction (XRD) shows only the characteristic (002) and (100) peaks for carbon, confirming the absence of bulk-like ordered metallic phases (Fig. S4). Figure 1c–e shows representative aberration corrected high-angle annular dark-field scanning transmission electron microscopy (AC-HAADF-STEM) images for the Fe, Rh, and La-N-C, where atomically dispersed metal sites are clearly observed through the high contrast points. Energy dispersive X-ray spectroscopy (EDS) maps confirm the homogenous distribution of carbon, nitrogen, and the relevant metal throughout the catalyst, Fig. 1c–e (see other M-N-Cs in Figs. S5–S8). Throughout the main text physical characterization of the Fe-, Rh- and La-N-C are shown as a 3d, 4d and f-metal representative for the set of M-N-C catalysts. Complete characterization for the remaining M-N-Cs is found in the ESI as noted.

The electron energy loss spectroscopy (EELS) in Fig. 2a–c shows the EELS spectra of small sample regions containing single atoms of Fe, Rh, and La, respectively. The EELS point spectra show the co-existence of the N K-edge and the corresponding metal-edge (Fe-L₃,₂, Rh-M₃, La-M₅,₄), supporting the bond formation between the singe metal atom and supporting nitrogen atoms (M-N_x). X-ray absorption spectroscopy (XAS) was used to further investigate the chemical state and coordination environment of the catalysts, including X-ray absorption near-edge structure (XANES) and extended X-ray absorption fine structure (EXAFS). The XANES edge position and features (e.g., the intensity of the while line) of the Fe K-edge (7112 eV), Rh K-edge (23,219.9 eV) and La L₃-edge (5890.6 eV) as compared to their corresponding reference foil and oxide compounds indicate oxidation states of *ca.* Fe²⁺, Rh³⁺, and La³⁺, in agreement with the Fe 2p, Rh 3d and La 3d X-ray photoelectron spectroscopy (XPS) analysis (Fig. S9). Interestingly, for Cr-N-C, evaluated to be one of the most selective M-N_x sites for the conversion of NO₃⁻ to NH₃ (Fig. 3d), the XANES analysis reveals a relatively high Cr oxidation sate, comprising a mixture Cr⁶⁺ (20–30%) and Cr³⁺ (70–80%). Analogously for other M-N-C catalysts, the chemical state of the metal center is assessed by their respective XANES and XPS spectra (Figs. S10–S16). Furthermore, the EELS valence state

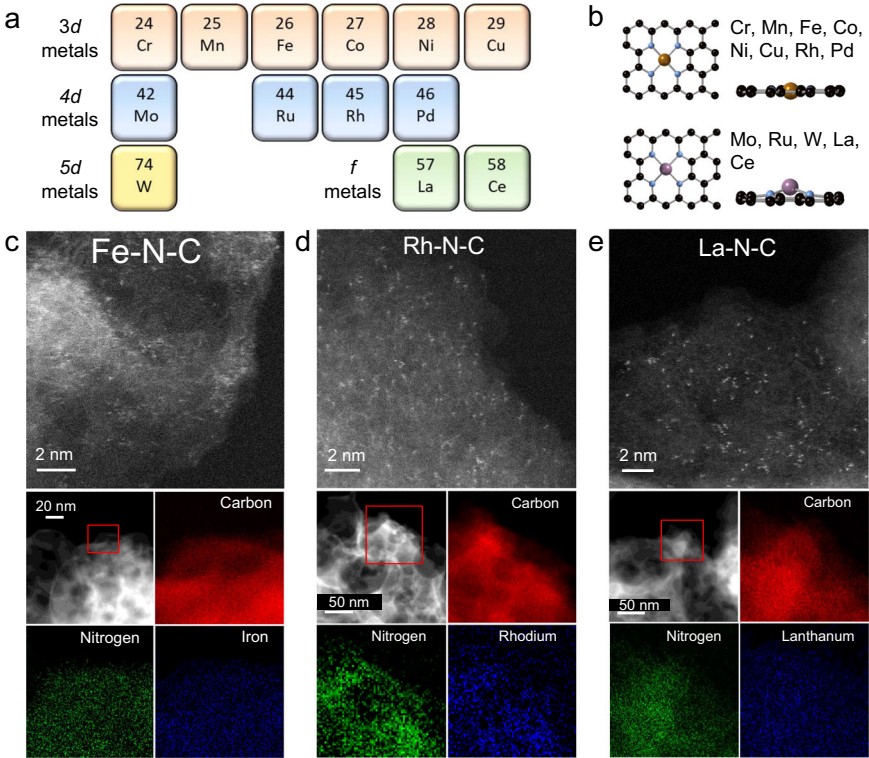

**Fig. 1 | Physical structure of the atomically dispersed transition-metal catalysts. a** Selection of *3d*, *4d*, *5d* and *f* metals synthesized via the sacrificial support method. **b** Schematic of the nitrogen coordinated metal active site (M-N$_4$) on a prototype carbon matrix, illustrating in-plane and out-of-plane configurations. Where the black, blue, brown/purple balls represent carbon, nitrogen and metal atoms, respectively. **c–e** Representative atomic resolution HAADF STEM images for the Fe-N-C, Rh-N-C and La-N-C catalysts. High contrast points indicate atomically dispersed metal sites. Corresponding EDS mapping of the M-N-C is given below.

analysis for Fe-N-C and La-N-C, evaluating the intensity ratio and separation distance between the Fe L$_3$/L$_2$ and La M$_5$/M$_4$ edges further supports the oxidation state of Fe (between Fe$^{2+}$ and Fe$^{3+}$) and La (La$^{3+}$) (Fig. S17). However, the Rh K-edge EELS signal is too weak to perform quantitative analysis. EXAFS provides structural information of the metal centers inner coordination spheres, confirming the presence of shorter bonds (M-C/M-N/M-O) rather than longer metallic M-M bonds. From comparison with complementary techniques (EELS and XPS) the formation of M-N bonds is supported. Fourier transformed (FT) EXAFS spectra for the Fe K-edge, Rh K-edge and La L$_3$-edge, respectively are shown in Fig. 2d–f. In the Fe-N-C and Rh-N-C spectra, a single sharp peak was observed at a bond distance of ca. 1.5 Å (phase uncorrected) for both, characteristic of the Fe-N and Rh-N coordination in the first shell. While for La-N-C, an *f*-metal with a complex coordination environment (shown to sit out of plane during the modeling of the active site, Fig. 1b) a peak was observed at higher distances between 1 and 2.5 Å (phase uncorrected). A similar peak can also be observed in the FT-EXAFS of amorphous La$_2$O$_3$. Therefore, we assume that in our La-N-C sample a strongly disordered local environment around the single atom site, including the possible formation of small metal oxide-clusters. For the Ce-N-C catalyst, a more complex peak split feature was observed indicating the complex nature of *f*-metal M-N-C catalysts (Fig. S18). To deconvolute minor contributions to the EXAFS spectra at larger bond distances, wavelet transforms of the EXAFS oscillations, providing high resolution information in both k-space and R-space are shown in Fig. 2g–I. For Fe-N-C, Rh-N-C, and La-N-C, in addition to the high intensity Fe-N, Rh-N and La-N peak, a low intensity peak is observed at *ca.* 3.5 Å, 3.2 Å, and 3.7 Å (at low k-space values), respectively which we attribute to Fe-C, Rh-C, and La-C interactions in the second coordination shell. The lack of peaks in wavelet-transformed EXAFS spectra at higher values of wavenumber *k* indicates the absence of metal-metal bonds in the M-N-C catalysts, and, hence, absence of

metallic or large oxide clusters. FT-EXAFS and WT-EXAFS for the M-N-C and corresponding metal oxide standards are given for Fe-, Rh-, and La-N-C in Fig. S19, while complete XANES, EXAFS, and FT-EXAFS spectra are provided for all M-N-Cs in Figs. S10–S12, with fitting parameters provided in Table S2 and fitting results plotted in R-space found in Fig. S20, respectively.

XPS was used to further examine the metal-nitrogen coordination and other nitrogen moieties present. The representative N 1s spectra for the Fe-, Rh-, and La-N-C catalysts in Fig. 2j–l, reveals the presence of metal-nitrogen moieties (M-N$_x$). N 1*s* XPS spectra for the remaining M-N-C catalysts is shown in Figs. S21–24, all of which demonstrate the formation of M-N$_x$ moieties. The deconvoluted high resolution N 1*s* XPS spectra showed a variation in the N-content and N-moiety percentage (e.g., pyridinic, pyrrolic, graphitic metal-N$_x$), as shown in Tables S3–6.

In summary, 13 atomically dispersed M-N-C catalysts have been synthesized using the SSM. By combining AC-STEM, EELS, XAS and XPS, the atomically dispersed nature of each metal site has been comprehensively visualized and confirmed to be in a M-N$_x$ coordination. Critically, the well-deciphered coordination environment of the metal center allows us to elucidate the intrinsic activity of the different M-N$_x$ moieties towards the electrocatalytic transformation of reactive N-species.

**Electrochemical performance**

The NO$_3$RR is somewhat universal in that even metal-free nitrogen-doped-carbon (N-C) showed limited but observable activity towards the generation of NH$_3$ and NO$_2^-$ under an isotopically labeled $^{15}$NO$_3^-$ feed in neutral electrolyte (pH = 6.3 ± 0.05) at −0.4 (V vs. RHE) (Fig. S25). Comparatively, with the addition of nitrogen-coordinated metal sites Cr-N-C as an example, the FE$_{NH3}$ increased from *ca.* 10% (metal free N-C) to 91.2 ± 9.6%. While the introduction of Fe-N$_x$ sites

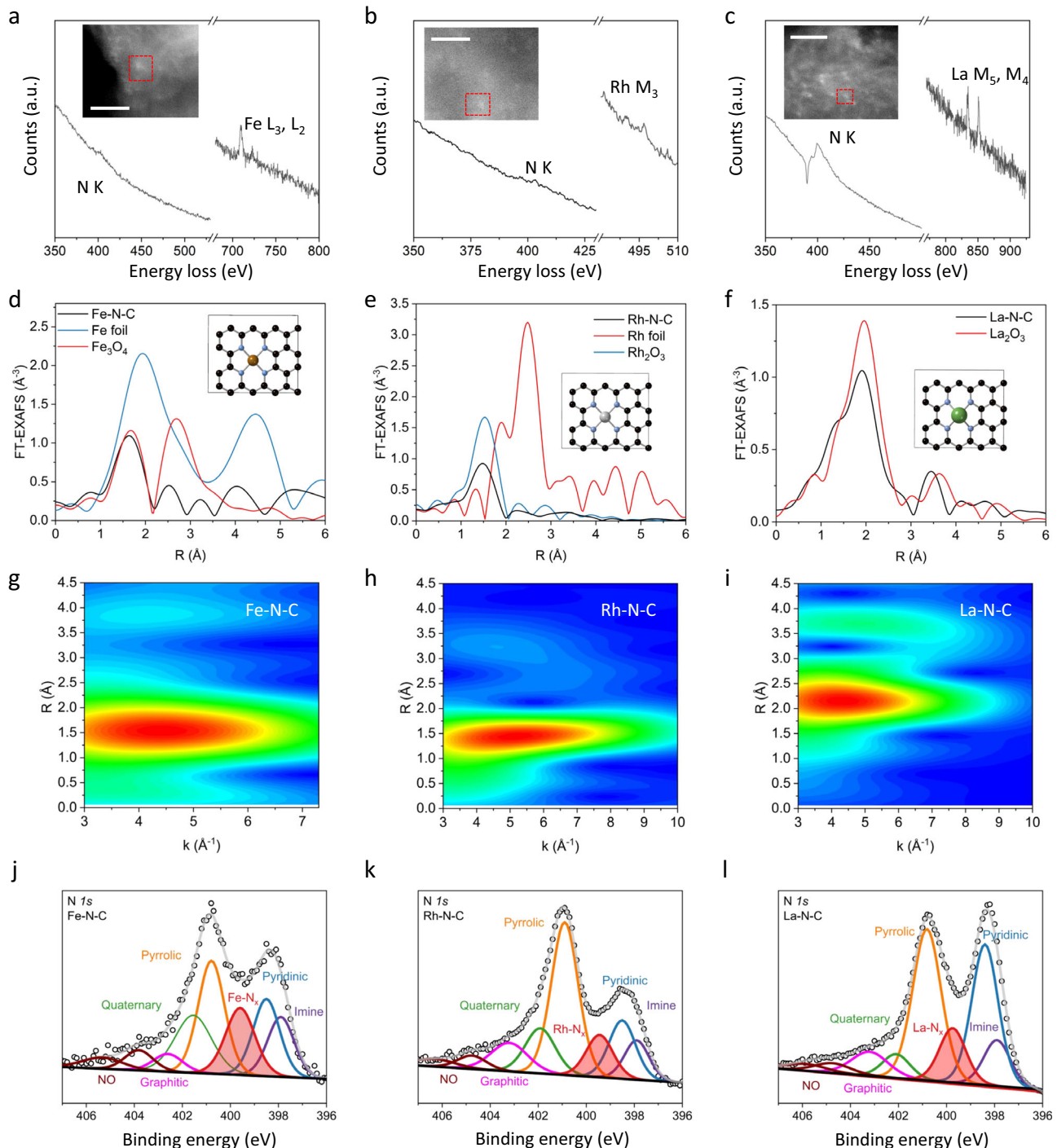

**Fig. 2 | Coordination environment of the single-atom metal centers. a–c** Atomic resolution EELS point spectra of the N K-edge and corresponding metal-edge of Fe ($L_{3,2}$), Rh ($M_3$) and La ($M_{5,4}$), in the M-N-C catalysts, indicating nitrogen coordination of the single atom M-$N_x$ site. All scale bars are 2 nm. **d–f** FT EXAFS spectra of the Fe, Rh and La-N-C catalysts, with the corresponding metallic foil and metal oxide standards, demonstrating the presence of M-$N_x$ sites. Where the black, blue, brown/ silver/green balls represent carbon, nitrogen and metal atoms, respectively. **g–i** WT-EXAFS of the Fe K-edge of Fe-N-C, Rh K-edge of Rh-N-C and La $L_3$-edge of La-N-C. **j–l** N 1s XPS spectra for the Fe, Rh and La-N-C catalysts, confirming the presence of M-$N_x$ moieties.

boosted the $FE_{NH3}$ to 99.1 ± 2.6% (again at −0.4 V vs. RHE) and the corresponding $Yield_{NH3}$ increased from 1.3 to 10.0 ± 1.3 μmol h$^{-1}$ cm$^{-2}$.

It should be noted that, for the $NO_3RR$ activity, as compared to measurements performed in neutral media, the current density and $NH_3$ yield rate could be significantly enhanced under alkaline conditions, while maintaining a high $FE_{NH3}$. Given Fe-N-C as an example, Fig. S26 shows that when using a 1 M KOH electrolyte (pH = 13.8 ± 0.17), the $NH_3$ partial current density increased from 2 mA cm$^{-2}$ (11.4 μmol h$^{-1}$ cm$^{-2}$) to 35 mA cm$^{-2}$ (156.5 μmol h$^{-1}$ cm$^{-2}$) at −0.4 V and further to 175 mA cm$^{-2}$ (750.4 μmol h$^{-1}$ cm$^{-2}$) at −0.6 V, both with a $FE_{NH3}$ above 95%. Other work on alkaline $NO_3RR$ also found similar trends[18,33,38]. However, as a fundamental study, this work focused on neutral pH (pH 6.3 ± 0.05) to evaluate the intrinsic activity of the M-$N_x$ sites toward the electrocatalytic $NO_3RR$ and $NO_2RR$.

Regardless of the $NO_3RR$ pathways[39–41], the surface intermediates *$NO_2$ and *NOOH are inevitable before the reaction pathway diverges

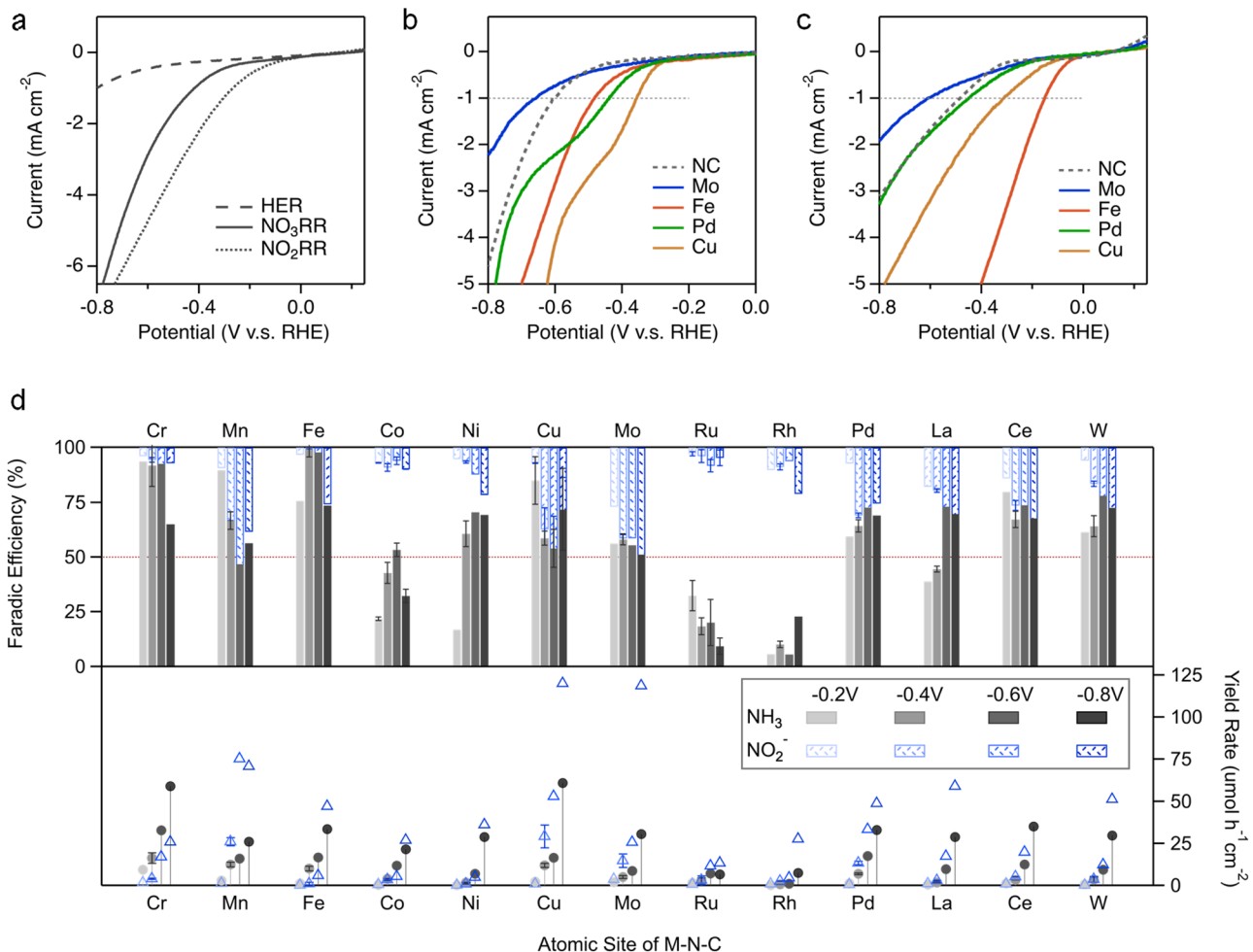

**Fig. 3 | Electrochemical activation, selectivity, and activity for the NO₃RR and NO₂RR. a** Example of a linear sweep voltammetry (LSV) curve for the Cr-N-C catalyst in electrolytes of 0.05 M PBS (HER), 0.05 M PBS + 0.01 M KNO₂ (NO₂RR) and 0.05 M PBS + 0.16 M KNO₃ (NO₃RR). Representative LSV over the metal free N-C and a variety of 3*d* and 4*d* metals for the (**b**) NO₃RR and (**c**) NO₂RR. Complete LSV for all examined M-N-C and metal free catalysts are given in Fig. S28. **d** Gap analysis plot (GAP) for the electrochemical NO₃RR for all M-N-C catalysts in 0.05 M PBS + 0.16 M KNO₃ for 2 h. The top section of the figure shows the Faradaic efficiency for NO₂⁻ (blue; top-down) and NH₃ (gray; bottom-up) as a function of the applied potentials between −0.2 V and −0.8 V vs. RHE. A horizontal line is set at 50% FE to guide the eye. The bottom section of the figure shows the corresponding yield rate (μmol h⁻¹ cm⁻²) for NO₂⁻ (blue; triangle) and NH₃ (gray; circle) as a function of the applied potentials. The corresponding NO₃RR chronoamperometry plots and UV–Vis detection for NO₂⁻ and NH₃ concentrations are shown in Figs. S29–38. Error bars are determined from three replicate trials at −0.40 V vs. RHE. The GAP for the metal free N-C catalysts is shown in Fig. S39.

to its several possible products. The NO₂⁻ molecule is also the first desorbable reaction intermediate, playing a key role in the NO₃RR via a cascade pathway[24,42]. For the set of M-N-C catalysts in this work, the NO₂RR displayed an earlier activation than the NO₃RR by linear sweep voltammetry (LSV), as shown in Fig. 3a, indicating that the 6e⁻ transfer from NO₂⁻/*NOOH to NH₃, in the NO₃RR was more facile than the initial 2e⁻ transfer from *NO₃⁻ to *NO₂⁻/*NOOH, which is the rate limiting step and thus the focus of the computational work in the following discussion. Representative LSV for the metal free N-C and a variety of 3*d* and 4*d* metals demonstrates the range of NO₃RR (Fig. 3b), and NO₂RR (Fig. 3c) activities, with Cu-N$_x$ being the most active NO₃RR site and Fe-N$_x$ being the most active NO₂RR site. As a unique nitrate-to-nitrite electrocatalyst[24], Mo-N-C was even more inert than the metal free N-C. A full analysis of the reaction onset potentials (Fig. S27) clearly visualizes the range of onset potentials for the diverse M-N$_x$ sites.

To investigate the NO₃RR activity and selectivity towards NH₃ and NO₂⁻, a series of 2-h potential holds were performed at potentials between −0.2 V and −0.8 V. A gap analysis plot (GAP) was constructed in Fig. 3d to visualize the nitrogen conversion landscape. Collectively, as the cathodic potential decreased, the yield rate of NH₃ and NO₂⁻ increased for all catalysts, but the selectivity varied. Specifically, both

Fe-N-C and Cr-N-C showed the highest NH₃ selectivity (FE$_{NH3}$ 92–100%) at −0.4 V and −0.6 V. In contrast, Mo-N-C showed comparable, consistent, and somewhat potential-independent selectivity for NO₂⁻ (40–50% FE$_{NO2}$) and NH₃ (50–55% FE$_{NH3}$), as expected given that Mo-N$_x$ sites were shown in our previous work, as an active nitrate-to-nitrite converter via dissociative adsorption (chemical process)[24]. Similarly, the Mn-, Pd-, W-, La-, and Ce-N-C showed moderate NO₂⁻ and NH₃ selectivity at high overpotentials (~70% FE$_{NH3}$ vs. ~30% FE$_{NO2}$) and the FE for nitrogen conversion (FE$_{NH3}$ + FE$_{NO2}$) was maintained at 100%. Other early transition metals (Co-, Ni- and Cu-N-C) showed irregular potential dependence on selectivity, wherein Co-N-C showed a volcano trend for FE$_{NH3}$ but consistent FE$_{NO2}$ and the nitrogen conversion was far below 100% FE through the entire potential range. For an easier comparison, the NO₃RR electrolysis results (FE$_{NH3/NO2}$ and Yield$_{NH3/NO2}$) have been separated based on the applied potential in Fig. S40, where its clearly observed that Cr-N-C was leading the other catalysts between −0.2 V and −0.6 V, being exceptional at −0.2 V (9.33 μmol h⁻¹ cm⁻² and 94% FE$_{NH3}$). However, Ni-N-C and Cu-N-C showed a strong potential dependence on the Yield$_{NH3}$, where Cu-N-C rivals Cr-N-C at −0.8 V (61 vs. 59 μmol h⁻¹ cm⁻²). Particularly, Cu-N-C maintained 100% FE for NH₃ + NO₂⁻ (FE$_{NH3}$ = 72%) at −0.8 V, indicating

that no current is wasted on the parasitic HER. Two distinct exceptions were Ru-N-C and Rh-N-C, albeit their early activation indicated by LSV (Fig. 3), both of which showed inferior $NH_3$ and $NO_2^-$ activity (Fig. 3d), displaying unique and significant gaps between the $NH_3$ and $NO_2^-$ FE bars. This gap in FE is likely due to insoluble, undetected gas phase products (Fig. S41), likely from the HER, as the isolated active sites present in atomically dispersed catalysts are unfavorable for the coupling of N-N bonds for $N_2$.

It was recently observed for the $NO_3RR$ and previously reported for the $CO_2RR$ that under a cathodic potential, atomically dispersed $Cu-N_x$ sites can reduce to form metallic clusters and even nanoparticles at more cathodic potentials[43–45]. Interestingly, additional investigations employing operando XAS over other atomically dispersed transition metal sites (M = Mn, Fe, Co, Ni and Pd) revealed a stable atomically dispersed $M-N_x$ site, even under highly reductive potentials[46–49]. Further studies employing post-mortem STEM observed atomically dispersed sites after reductive potentials in the $CO_2$ and $N_2$ reduction reactions, over Ru-, Rh-, Ce-N-C, and other rare earth metals[50–54]. To investigate possible morphological changes to the atomically dispersed nature of the catalysts in this work after the $NO_3RR$ electrolysis, the representative Fe-, Rh-, and La-N-C catalysts were imaged. Post-mortem STEM images in Figs. S42–S44 confirm atomically dispersed nature of the Fe-, Rh- and $La-N_x$ sites after the series of $NO_3RR$ electrolysis from −0.2 to −0.8 V for 2 h each. It is important to note that post-mortem STEM does not reveal possible in-situ restructuring under electrolysis conditions. However, as reported in a recent study employing Cu-N-C for the $NO_3RR$, as morphological reconstructions were occurring as a result of the cathodic potential, a significant increase in the current is readily observed over time in the electrochemical response (chronoamperometry curves) as the $Cu-N_x$ sites transformed into metallic Cu clusters/particles[43]. Similar changes in the electrochemical response would be expected in the current study during both the $NO_3RR$ and $NO_2RR$ if significant morphological reconstruction was occurring, however, this was not observed. This may suggest, however not confirm without operando XAS, that even for Cu-N-C (which has been shown to be more susceptible to morphological changes under reductive potentials than other M-N-C catalysts), significant morphological reconstruction of the $Cu-N_x$ sites may not be occurring. Perhaps because of variations in the stability of the $Cu-N_x$ sites, originating from the different M-N-C synthesis approaches and precursor selections. This behavior is observed in the literature for Cu-N-C, by comparing operando studies under reductive potentials, for which some studies report active site reconstruction at −0.2 V vs. RHE[43] and others report no changes up to −0.6 V vs. RHE[44]. Additionally, at the mild cathodic potential in this work of −0.2 V, at which the correlations (discussed in the following section) are most accurate, significant morphological reconstruction may not be expected and was not observed over time in the electrochemical $NO_3RR$ and $NO_2RR$ performances, however, direct spectroscopic evidence would be required to confirm this hypothesis. Although no $M-N_x$ restructuring was observed through post-mortem STEM (Fe-, Rh-, and La-N-C) or suggested by the electrochemical response, this cannot be totally ruled out and will require future operando XAS studies for a robust evaluation.

For the role of potential nitrite (as a by-product or intermediate), previous reports usually considered the $NO_3RR$ as a direct $8e^-$ transfer pathway with certain irreversible $NO_2^-$ desorption or leaching[32,33,55]. Here, doping of isotopic $^{15}NO_2^-$ in the $NO_3RR$, schematically shown in Fig. 4a, revealed that trace amounts (e.g., 1 ppm) of $^{15}NO_2^-$ could be easily reduced to $^{15}NH_3$ even under a concentrated $^{14}NO_3^-$ environment (10,000 ppm), which applied to both $M-N_x$ sites and metal-free N-C sites (Figs. S45–S46). Figure 4b shows the NMR spectra for a time course electrolysis over Fe-N-C, wherein even at a concentration ratio of 1,000:1 ($^{14}NO_3^-$ : $^{15}NO_2^-$), the $^{15}NO_2^-$ competed for an active site, yielding comparable $^{15}NH_3$. NMR spectra for the other M-N-C catalysts is shown

in Fig. S47, where a similar trend is observed for most metals with the no exception of Co- and Ni-N-C, where significantly more isotopic $^{15}NH_3$ is generated than standard $^{14}NH_3$, in support of Fig. 3d, suggesting their poor activity towards $NO_3^-$ but high activity towards $NO_2^-$, which will be addressed in the following section. As the concentration ratio is reduced to 100:1 ($^{14}NO_3^-$ : $^{15}NO_2^-$), the $^{15}NO_2^-$ outcompeted $^{14}NO_3^-$, yielding significantly more $^{15}NH_3$ than $^{14}NH_3$ over the 6-h electrolysis (Fig. 4c). This indicates that for the nitrogenous products in the $NO_3RR$, the potential participation of bulk $NO_2^-$ or locally channeled $NO_2^-$, significantly complicates the mechanism of $NH_3$ production (e.g., $2e^- + 6e^-$ vs. $8e^-$)[24,56]. Therefore, unambiguously identifying whether the underlying reaction mechanism is a direct $8e^-$ pathway, a (rapid) $2e^- + 6e^-$ pathway with a bulk/channeled $NO_2^-$ intermediate, or a combination thereof, is of great significance for the design and optimization of $NO_3RR$ systems and even more complex nitrate-involving reactions[57,58].

Figure 4d (and shown individually by applied potential in Figs. S48–S49) shows the $NO_2RR$ electrolysis for all M-N-C catalyst under a 0.01 M $NO_2^-$ feed, a concentration mimicking the bulk $NO_2^-$ concentrations in the $NO_3RR$ electrolysis (Fig. 3d). Obviously, both the $NH_3$ selectivity and activity of the $NO_2RR$ were significantly higher than that of the $NO_3RR$ (Fig. 3d) for all catalysts. Specifically, Fe-, Co-, and La-N-C showed 100% $FE_{NH_3}$ over the whole potential range and Cr-, Ni-, Cu-, Pd-, and W-N-C showed increasing $FE_{NH_3}$ as the cathodic potentials decreased, reaching 100% at −0.8 V. Again, Mo-N-C showed a unique and potential-independent $FE_{NH_3}$ around 75%. Similar to the $NO_3RR$, the $NO_2RR$ for Ru-N-C and Rh-N-C showed a distinct decreasing trend on the $FE_{NH_3}$ and consistent $Yield_{NH_3}$ over the entire potential range, likely being outcompeted by the HER as the cathodic potential decreased.

To examine the relationship between the $NO_2RR$ and $NO_3RR$, Fig. 4e, f correlates the activity of $NO_2RR$ ($Yield_{NH_3}/FE_{NH_3}$) with the selectivity of $NO_3RR$ ($FE_{NH_3}$), revealing a linear relationship at −0.20 V and −0.40 V. This linear relationship suggests a major contribution from the bulk or locally channeled $NO_2^-$ towards $NH_3$ ($2e^- + 6e^-$ pathway). Meanwhile, the correlation between the $FE_{NO_2-}$ in the $NO_3RR$ and the $Yield_{NH_3}$ in the $NO_2RR$, has a poor linear fit (Fig. S59), highlighting the complex reaction mechanism. In contrast, the poor correlations between the $NO_3RR$ $FE_{NO_2-}$ and $NO_2RR$ $Yield_{NH_3}$ (Fig. 4g), confirmed that in the $NO_3RR$ the bulk $NO_2^-$ species were in a complex production-consumption process rather than an irreversibly desorbed final by-product.

## Computational descriptors for the $NO_3RR$ and $NO_2RR$

As discussed above, the first $2e^-$ transfer is the rate limiting step in the $NO_3RR$, therefore DFT was used to study the energetics of key intermediates in the conversion of $NO_3^-$-to-$NO_2^-$ on various catalytic active sites (Fig. S61). This included the $^*NO_3$ surface intermediate formed by oxidative associative adsorption of $NO_3^-$, the $^*O$ surface intermediate formed by neutral dissociative adsorption of $NO_3^-$[24], and the $^*NO_2$ surface species formed in the reductive adsorption of $NO_3^-$, these result in four thermodynamic reaction descriptors, as shown in Fig. 5a.

Figure 5a shows that the Mo-$N_4$, La-$N_4$, Ce-$N_4$, and W-$N_4$ sites could strongly adsorb $NO_3^-$ ($\Delta_rG \sim -4eV$) in a dissociative manner, forming an $^*O$ surface species and a free $NO_2^-$ molecule. For these oxyphilic metals, upon exposure to $NO_3^-$ molecules, the single-atom sites are oxygenated with a fifth ligand and the new active site (O-M-$N_4$) can still coordinate a $NO_3^-$/$NO_2^-$ molecule for further reduction[24]. Figure S62 shows that the O-M-$N_4$ sites had a weaker interaction with $NO_3^-$ but could stabilize the $^*NO_3$/$^*NO_2$ surface intermediates. For the Cr-$N_4$, Mn-$N_4$, Fe-$N_4$ and Ru-$N_4$ sites, associative and dissociative adsorption of $NO_3^-$ is likely a competitive process that also depends on the cell potential. Namely, while the associative adsorption of $NO_3^-$ is potential dependent, dissociative adsorption is not as no electrons are involved

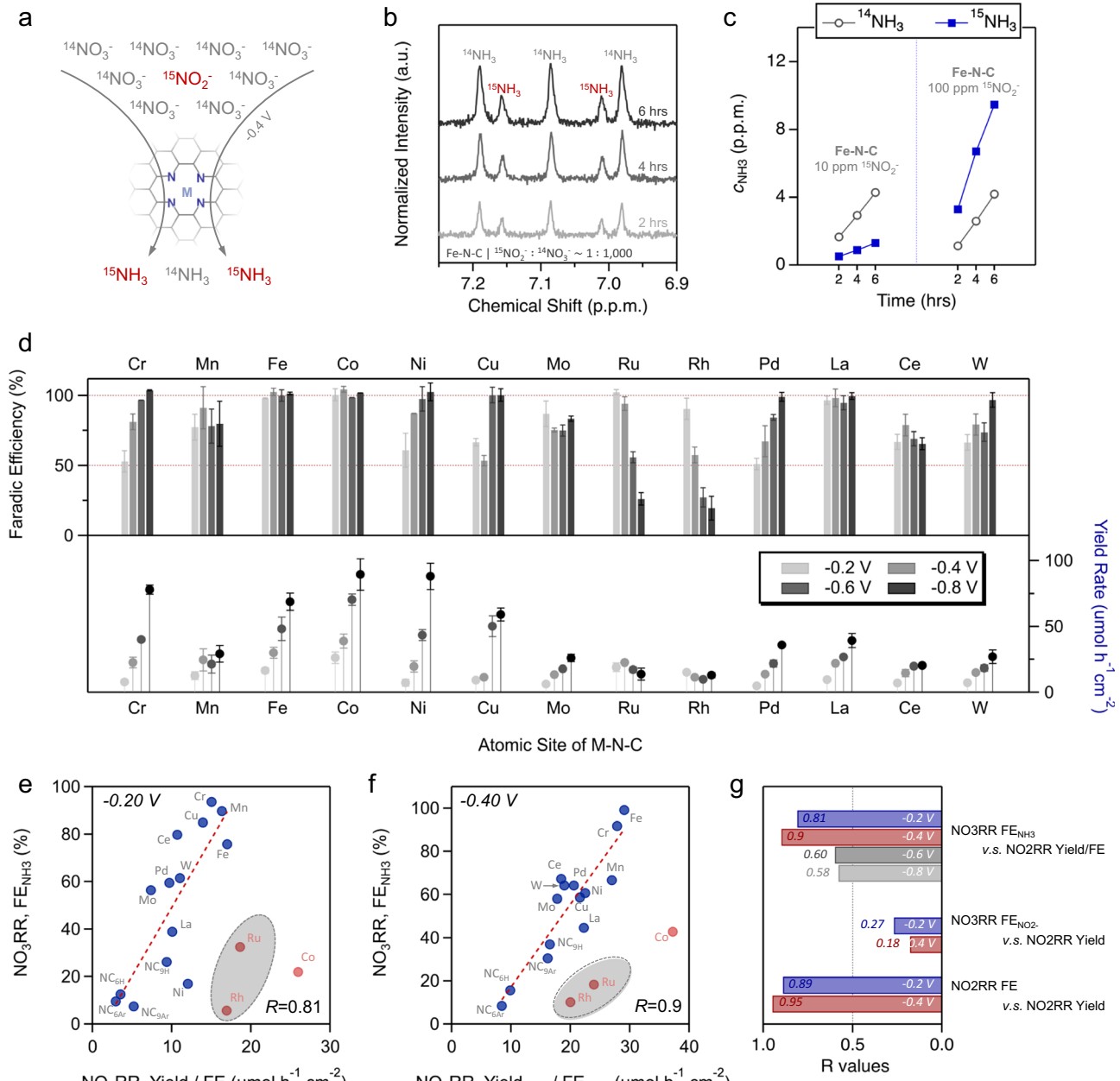

**Fig. 4 | Mechanistic analysis of the NO₃RR via a NO₂⁻ intermediate, NO₂RR performance and experimental correlations between lining NO₃RR selectivity and NO₂RR performance. a–c** Isotopic analysis of competing NO₃RR and NO₂RR reactions at −0.4 V. **a** Schematic for NO₃RR electrolysis in which small amounts of $^{15}NO_2^-$ are doped in 0.16 M $^{14}NO_3^-$. **b** NMR spectra for the electrolysis with 10 ppm of $^{15}NO_2^-$ doped in 0.16 M $^{14}NO_3^-$ (10,000 ppm), sampled at 2, 4, and 6 h. **c** Time course analysis of $^{14}NH_3$ and $^{15}NH_3$ concentration at 2, 4, and 6 h for the NO₃RR with 10 ppm $^{15}NO_2^-$ (left) and 100 ppm $^{15}NO_2^-$ (right) in 0.16 M $^{14}NO_3^-$ (10,000 ppm). **d** Electrochemical NO₂RR for M-N-C catalysts in 0.05 M PBS + 0.01 M KNO₂ for 0.5 h. Faradaic efficiency for NH₃ as a function of the applied potential between −0.2 V and −0.8 V (top). Bottom section shows the corresponding NH₃ yield rate. The chronoamperometry plots and UV-Vis detection curves for NH₃ are

shown in Figs. S50–56. Note, FE's over 100% arise from electrolyte dilution to bring the concentration of NH₃ into the UV-vis calibrated range. Error bars are determined from three replicate trials. Correlations between the NO₃RR Faradaic efficiency for NH₃ in Fig. 3d, where *R* is the correlation coefficient (y-axis) and the dividend of the NH₃ yield rate and Faradaic efficiency of the NO₂RR in Fig. 4d at (**e**) −0.20 V and (**f**) −0.40 V. The NO₂RR, Yield$_{NH3}$ /FE$_{NH3}$ is represented as the NO₂RR total current (NO₂RR, j$_{total}$). The correlation between the NO₂RR, Yield$_{NH3}$ and NO₃RR, FE$_{NH3}$ is shown in Fig. S57. Ru and Rh outliers (gray shades) were excluded from the linear fit due to the dominant NO₃RR gaseous products as shown in Fig. S41 as well as Co-N-C. **g** Summary of the correlation coefficients, *R*, for varying experimental NO₃RR-NO₂RR correlations. Correlations not shown in Fig. 4e, f, are shown graphically in Figs. S58–60.

in this process. Co-N₄, Ni-N₄, Cu-N₄, Pd-N₄, and La-N₄ sites adsorb NO₃⁻ associatively and produce NH₃ via a *NO₂ intermediate. Additionally, our DFT descriptors suggest limited NO₃RR activity over Rh-N₄ sites, being only favorable via the reductive adsorption of NO₃⁻, while in other computational based works, Rh-N₄ sites suffer some competition between *H and *NO₃, in agreement with the poor NO₃RR activity observed in our experimental observations[41,59].

The quadrant plot in Fig. 5b graphically shows the correlations between the DFT-derived descriptors, mapping out the competitive NO₃⁻ adsorption and sources for NO₂⁻ evolution. Specifically, the far ends of Quadrant I, Quadrant II and Quadrant IV.b represent unfavored NO₃⁻ adsorption, exclusive NO₂⁻ evolution and unfavored NO₂⁻ evolution, respectively. These three blank sectors explain the above-mentioned universal NO₃RR activity for all M-N-C and metal free N-C

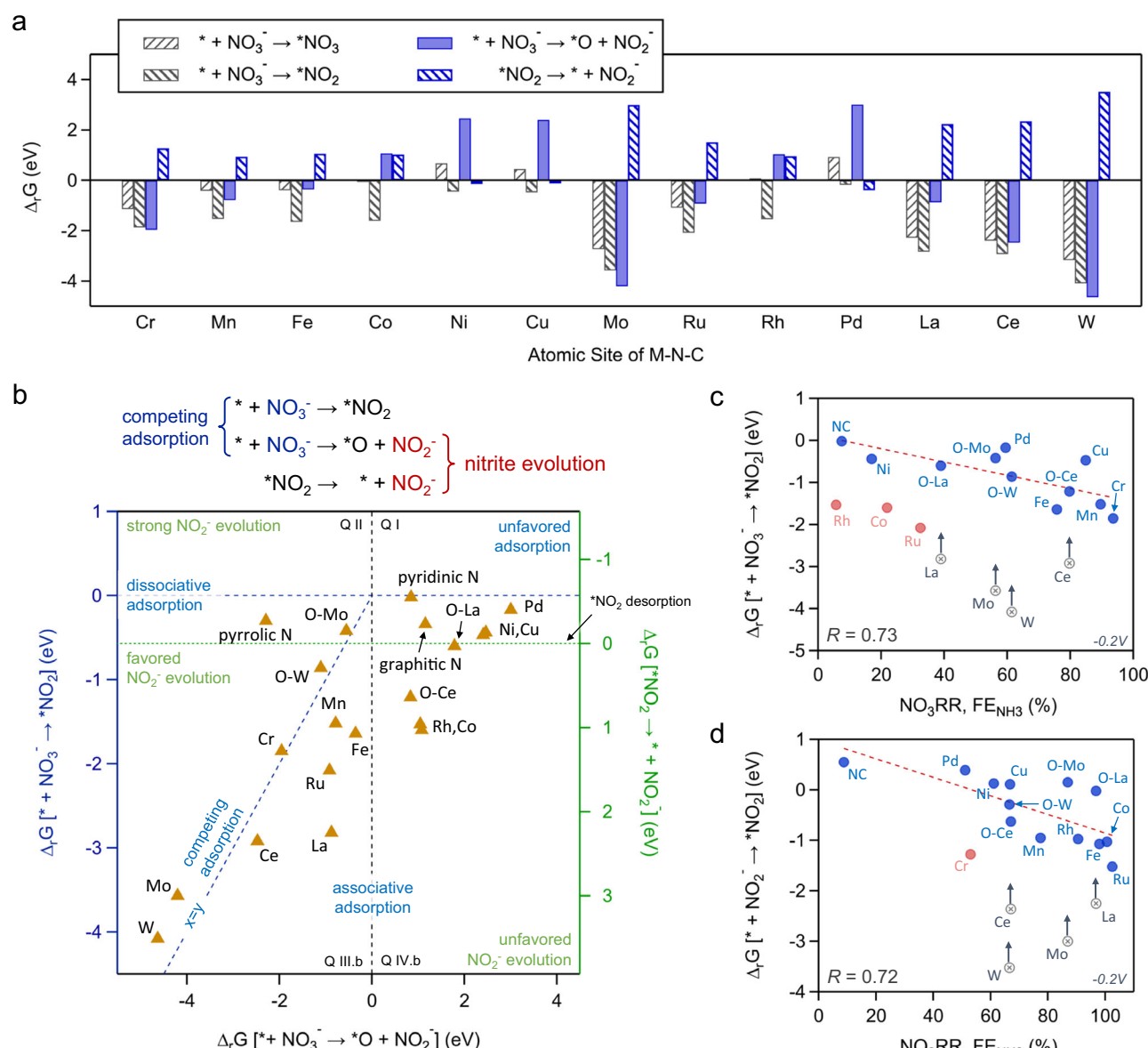

**Fig. 5 | Computational NO₃RR descriptors as calculated by using DFT with optB86b-vdW functional. a** Gibbs free energies ($\Delta_rG$) of the reaction for the first two electron transfer steps in the NO₃RR. Processes that generate NO₂⁻ in the bulk electrolyte are shown in blue. Note solid bars indicate no electron transfer, uphill gradient stripes indicate a reductive e⁻ transfer and downhill gradient stripes indicate an oxidative/reverse electron transfer. **b** Quadrant plot of the $\Delta_rG$ for associative adsorption [* + NO₃⁻ → *NO₂] (Y₁-axis), dissociative adsorption [* + NO₃⁻ → *O + NO₂⁻] (X-axis) and *NO₂ desorption [*NO₂ → * + NO₂⁻] (Y₂-axis), forming sectors where certain reaction pathways are thermodynamically favored. The main quadrants were determined by the X-axis and Y₁-axis. Quadrants III and IV

were further divided by the Y₂-axis into III.a, III.b, IV.a and IV.b sub‑sections. The three reaction coordinates determined two types of NO₃⁻ adsorptions and two types of NO₂⁻ evolutions as shown by the diagram above the figure. For simplified plots correlating two descriptors at a time, see Fig. S63. **c** Correlation between DFT-derived $\Delta_rG$[* + NO₃⁻ → *NO₂] and experiment-derived NO₃RR $FE_{NH3}$ at −0.2 V vs. RHE. **d** Correlation between DFT-derived $\Delta_rG$[* + NO₂⁻ → *NO₂] and experiment-derived NO₂RR $FE_{NH3}$ at −0.2 V vs. RHE. Oxygenated active sites (O-M) were included for the oxyphilic elements (Mo, La, Ce and W), where $R$ is reported as the absolute value of the correlation coefficient.

catalysts (Fig. S64), as well as the variant but non-dominant FE$_{NO2}$ in the NO₃RR. The diagonal of Quadrant III (x = y) indicates competitive associative and dissociative adsorption of NO₃⁻. The largest NO₂⁻ producing catalyst, Mo-N-C (O-Mo-N₄ and Mo-N₄ sites), is located at the diagonal, indicating comparable associative and dissociative adsorption. Several metal centers (e.g., Mn, Fe, Ru, La, and Ce) fall below the diagonal in Quadrant III.b and upper Quadrant IV.b, suggesting a favored associative NO₃⁻ adsorption for high NH₃ selectivity in NO₃RR electrolysis (Fig. 3d). Quadrant IV.b (Co, Rh, O-Ce and O-La) marks the region where a solely direct 8e⁻ pathway to NH₃ is feasible, with formation of the *NO₂ intermediate being favored, while both descriptors for the generation/desorption of NO₂⁻ are unfavored.

Quadrant IV.a (Ni, Cu and Pd, pyridinic-N and graphitic-N) shows a weak stabilization of the *NO₂ intermediate, while simultaneously showing activity for the desorption of *NO₂ for NO₂⁻ evolution. It should be noted that Mo-N-C is the only metal center favoring NO₂⁻ evolution through both the dissociative NO₃⁻ adsorption and *NO₂ desorption paths, explaining the unique selectivity of Mo-N-C in the NO₃RR (Fig. 3d). The DFT calculated Gibbs free energy changes for these descriptors are given in Table S7.

To evaluate the practical relevance of these computational descriptors, a set of correlations were developed between the DFT-derived free energies ($\Delta_rG$) in Fig. 5a and the electrocatalytic performance in Figs. 3d and 4d. Specifically, Fig. 5c shows the correlation

between the adsorption energy of $NO_3^-$ ($\Delta_r G[^* + NO_3^- \rightarrow {}^*NO_2]$) and the $NO_3RR$ $NH_3$ selectivity ($FE_{NH3}$), wherein a linear correlation ($R = 0.73$) is observed at $-0.2$ V. This result highlights the importance of a stable $^*NO_2$ intermediate for high $NH_3$ selectivity in both the $8e^-$ pathway or the $2e^- + 6e^-$ pathway (which requires the re-adsorption of $NO_2^-$). Additionally, the involvement of the oxo-form for the highly oxyphilic elements (O-Mo, O-W, O-La, O-Ce) shifts them back onto the trend line as compared to their bare $M-N_4$ counterparts. These results suggest the distinct active sites/reaction mechanism of early transition $4d$-, $5d$- and $f$-metal-based M-N-C catalysts in neutral environments, which could be attributed to the oxophilicity and large coordination number of these metals that allow simultaneous coordination of multiple intermediates[60]. Similarly, Fig. 5d shows a comparable correlation ($R = 0.72$) between the adsorption energy of $NO_2^-$ ($\Delta_r G[^* + NO_2^- \rightarrow {}^*NO_2]$) and the $NO_2RR$ ammonia selectivity ($FE_{NH3}$), indicating that the stabilization of the $^*NO_2$ intermediate also plays a key role in the $NO_2RR$ as well as the downstream $6e^-$ transfer in the $NO_3RR$. It is important to note potential limitations of these computational-experimental correlations. From our observations, metal centers over which H* adsorption is a competing factor leading to the HER dominating (Ru- and Rh-N-C, Fig. 3) the $NO_3RR$ are not well predicted by these correlations. Similarly, metal centers which demonstrate significant $NO_2^-$ activation, with poor $NO_3^-$ activation (observed over Co-N-C in the isotopic $NO_2^-$ doping experiments, Fig. S47) are also not well predicted by these correlations.

However, the $NO_3RR$ $FE_{NO2^-}$ showed minimum correlations with either $\Delta_r G[^*NO_2 \rightarrow {}^* + NO_2^-]$ or $\Delta_r G[^* + NO_3^- \rightarrow {}^*O + NO_2^-]$, as shown in Fig. S65. This agrees well with the above-mentioned poor correlation between the $NO_3RR$ $FE_{NO2^-}$ and $NO_2RR$ $Yield_{NH3}$ (Figs. 4g and S59), wherein the bulk $NO_2^-$ species in the $NO_3RR$ were in an active but and complex production-consumption process, making it difficult to deconvolute the $NO_2^-$ production and consumption rates from its net yield.

## Discussion

In summary, a rich set of atomically dispersed $3d$-, $4d$-, $5d$-, and $f$-block M-N-C catalysts with a well-established $M-N_x$ coordination was synthesized. The gap analysis plot revealed diverse $NO_3RR$ performance, wherein Cr-N-C and Fe-N-C were the most $NH_3$ selective catalysts, achieving near 100% $FE_{NH3}$, while Mn-, Cu-, and Mo-N-C were highly selective for $NO_2^-$. For the $NO_2RR$, several elements including Cr-, Fe-, Co-, Ni-, Cu- and La-N-C achieved a $FE_{NH3}$ of 100% at high overpotentials, with Fe and Co-N-C showing 100% $FE_{NH3}$ over the entire potential range. Isotopically doped $^{15}NO_2^-$ in concentrated $^{14}NO_3^-$, demonstrated the ease at which minute concentrations of $NO_2^-$ in the bulk electrolyte can preferentially reduce to $NH_3$, convoluting the possibility of a direct $8e^-$ pathway or a $2e^- + 6e^-$ cascade pathway with $NO_2^-$ as transient intermediate for the $NO_3RR$. The correlation between experimental $NO_3RR$ ammonia selectivity and experimental $NO_2RR$ activity suggested a universal contribution of the $NO_2^-$ intermediate in $NO_3RR$ ($2e^- + 6e^-$). The DFT-derived thermodynamic descriptors can theoretically explain the electrocatalytic selectivity for each metal center. Furthermore, these computational descriptors showed good correlations with the experimental performances, such that a simple computationally evaluated descriptor can be utilized to estimate the activity of M-N-C catalysts for the $NO_3RR$ and $NO_2RR$. While these computational-experimental activity descriptors provide correlations for the $NO_3RR$ and $NO_2RR$ activity, these correlations are limited when the $H^+$ adsorption is a competing factor leading to the HER over $NO_3RR$ (Ru- and Rh-N-C) and when there is poor $NO_3^-$ activation in contrast with significant $NO_2^-$ activation (Co-N-C). Importantly, these computational-experimental activity descriptors are based on the $M-N_x$ active site and have strong predictive $NO_3/NO_2RR$ ability at low potentials, where the intrinsic nature of the $M-N_x$ site is observed. However, the effectiveness of these activity descriptors decreases as

the overpotential increases, forcibly driving the reaction and possibly inducing structural changes to the $M-N_x$ sites, convoluting the intrinsic activity of the atomically dispersed metal center. This work deciphered the unique fundamentals of the $NO_3RR$ over atomically dispersed M-N-C catalysts to create a set of experimentally driven computational descriptors for the $NO_3/NO_2RR$ activity, paving the road for the design of tandem $NO_3RR$ systems and nitrate-containing systems composed of either multi-metallic M-N-C catalysts or extended catalytic surfaces supported by synergistic M-N-C supports.

## Methods

### Materials

Nicarbazin (Sigma-Aldrich), CAB-O-SIL® LM-150 fumed silica (Cabot), Aerosil® OX-50 (Evonik), iron(III) nitrate nonahydrate (Sigma-Aldrich), chromium(III) acetylacetonate (Sigma-Aldrich), manganese(II) nitrate tetrahydrate (Sigma-Aldrich), cobalt(II) nitrate hexahydrate (Sigma-Aldrich), nickel(II) nitrate hexahydrate (Sigma-Aldrich), copper(II) nitrate hemi pentahydrate (Sigma-Aldrich), ammonium molybdate tetrahydrate (Sigma-Aldrich), ruthenium(III) nitrosylnitrate (Alfa Aesar), rhodium(III) nitrate hydrate (Sigma-Aldrich), palladium(II) nitrate dihydrate (Sigma-Aldrich), lanthanum(III) nitrate hexahydrate (Alfa Aesar), cerium(III) nitrate hexahydrate (Fisher Scientific) and ammonium paratungstate (Sigma-Aldrich), potassium nitrate (Sigma-Aldrich), potassium nitrite (Sigma-Aldrich), isotopic potassium nitrate ($^{15}N$, 99% - Cambridge Isotope Laboratories), isotopic sodium nitrite ($^{15}N$, 98%⁺ - Cambridge Isotope Laboratories).

### Synthesis of atomically dispersed M-N-C catalysts

All catalysts were synthesized following the well-established sacrificial support method, as detailed below, with the metal precursor loading, pyrolysis temperature, pyrolysis atmosphere and etching environment being tuned to maintain atomically dispersed metal sites between the different metals (Supplementary note 1).

**Synthesis of Mn, Fe, Co, Ni, Cu, Mo and W-N-C catalysts.** First a slurry of a carbon-nitrogen containing precursor, Nicarbazin (6.25 g), the silica sacrificial support, LM-150 (Cabot, 1.25 g), OX-50 (Evonik, 1.25 g) and Stöber spheres (made in house, 0.5 g) and the corresponding metal salt precursor (Mn = 0.266 g, Fe = 0.60 g, Co = 0.272 g, Ni = 0.271 g, Cu = 0.345 g, Mo = 0.262 g and W = 0.095 g), in 50 mL of MilliQ water was created. Next, the slurry was sonicated for 30 min before being dried overnight at 45 °C, under constant stirring. The mixture was then further dried in an oven at 45 °C for 24 h. The resulting powder was then ball milled at 45 Hz for 1 h. The catalyst powder is then pyrolyzed at 975 °C (with a ramp rate of 15 °C min⁻¹) under a reductive $H_2/Ar$ (7%/93%) atmosphere for 45 min. The pyrolyzed powder is then ball milled a second time at 45 Hz for 1 h. The silicate template is then etched in a hydrofluoric acid (15 M) solution for 96 h. The catalyst is then recovered by filtration and washed to neutral pH, followed by drying at 60 °C overnight. The catalyst undergoes a second pyrolysis in a reductive $NH_3/N_2$ (10%/90%) atmosphere at 950 °C for 30 min (with a ramp rate of 20 °C min⁻¹). The catalyst is then ball milled a final time at 45 Hz for 1 h.

**Synthesis of Cr-N-C catalyst.** The synthesis is identical to the previous procedure, with only the pyrolysis temperatures being reduced to 650 °C in both the first and second pyrolysis, to maintain an atomic dispersion of the Cr metal. The metal salt precursor loading of Cr = 0.519 g.

**Synthesis of Ru-N-C catalyst.** The synthesis is identical to the previous procedure, with the pyrolysis temperatures being reduced to 650 °C and an inert argon pyrolysis atmosphere, to maintain an atomic dispersion of the Ru metal. The metal salt precursor loading of Ru = 0.235 g.

**Synthesis of Rh and Pd-N-C catalysts.** The synthesis is identical to the previous procedure, with the high pyrolysis temperatures of 975 °C and 950 °C for the first and second pyrolysis, respectively, however the pyrolysis is performed in an inert argon atmosphere, to maintain an atomic dispersion of the Rh and Pd metals. The metal salt precursor loading of Rh = 0.215 g and for Pd = 0.198 g.

**Synthesis of the La and Ce-N-C catalysts.** The synthesis is identical to the previous procedure, with the pyrolysis temperatures being reduced to 650 °C under an inert argon pyrolysis atmosphere. Furthermore, the etching of the silica template was performed in an alkaline 4 M NaOH environment at 80 °C for 96 h. If etched using a hydrofluoric acid environment, the La and Ce readily form $LaF_3$ and $CeF_3$ nanoparticles. The metal salt precursor loading of La = 0.060 g and Ce = 0.061 g.

## Physical characterization
The physical structure and atomically dispersed nature of the catalysts were analyzed by aberration-corrected scanning transmission electron microscopy (AC-STEM) and EDS using a JEOL ARM300CF at an accelerating voltage of 300 kV. The coordination environment and valence state of the atomically dispersed sites were examined through atomic resolution electron energy loss spectroscopy (EELS) on a Nion Ultra-STEM200 microscope equipped with a cold FEG, a C3/C5 aberration correction and a high-energy resolution monochromated EELS system (HERMES). To mitigate the beam damage on the coordination between single metal atoms and nitrogen-carbon support, the EELS was collected at a low voltage of 60 kV. During acquisition, the energy dispersion was set as 0.16 ~ 0.3 eV/channel with an exposure time of 2–4 s nm$^{-2}$. The background in each spectrum was removed by a power-law function and the de-noising of the spectra was performed by the multivariate weighted principal component analysis routine in the Digital Micrograph software.

The hierarchical pore structure of the catalyst was examined by scanning electron microscopy (SEM) on an FEI Magellan 400 XHR SEM.

The electronic structure and local bonding environment of the metal sites were analyzed by ex-situ X-ray Absorption Spectroscopy (XAS) collected on several beamlines. Rh K-edge was measured on the SAMBA beamline at SOLEIL synchrotron radiation facility, Paris, France. The sample was measured in fluorescence mode and references in transmission mode using a Si (220) monochromator for the energy selection. Ionization chambers to measure the X-ray intensity before and after the sample were filled with a mixture of Ar/N$_2$ ($I_0$) or pure Ar ($I_1$/$I_2$).

The La L$_3$-edge was measured on the XAFS beamline at ELETTRA synchrotron radiation facility, Triest, Italy. The sample was measured in transmission mode using a Si (111) 20% detuned monochromator for the energy selection. Simultaneously, a vanadium reference foil was measured for the energy calibration. Ionization chambers to measure the X-ray intensity before and after the sample were filled with a mixture of N$_2$/He ($I_0$, $I_1$, $I_2$).

The Fe K-edge was measured on the lab-based easyXAFS300 at the Fritz Haber Institute, Berlin, Germany (measured in transmission mode). Using a Ge (620) crystal for the energy selection and a Si drift detector (AXAS-M assembly from KETEK GmbH). A W-ProtoXRD X-ray tube was used.

The Ce L$_3$-edge, Cr, Mn, Co, Ni and Cu K-edge were measured on the KMC-3 beamline at the Bessy synchrotron radiation facility, Berlin, Germany. Cr-N-C in transmission mode, Cu-, Co-, Ni-, Mn- and Ce-N-C in fluorescence mode using a Si (111) monochromator for the energy selection. The ionization chamber for the measurement of the X-ray intensity before the sample was filled with 100 mbar air ($I_0$). Either a 13 element Si drift detector was used (fluorescence) or PIPS detector (transmission). Note that the energy resolution of the XAS spectra measured at the KMC-3 beamline is compromised due to

technical issues involving the optics/mirrors at the KMC-3 beamline. However, the samples and corresponding reference foils and compounds were measured for energy calibration and comparison of features. Note Mo-N-C and Pd-N-C were measured on the 10-BM beamline of the Advanced Photon Source at Argonne National Laboratory.

The ATHENA software was used for data alignment and XAS spectra extraction[61]. A set of in-house built Wolfram Mathematica scripts were used for the XANES analysis. Fitting of the EXAFS spectra was done using FEFFIT scripts[61]. The photoelectron amplitudes and phases were calculated by the FEFF8 code[62]. The EXAFS R-space fitting parameters and results are given in Table S8. If the M-N-C sample and corresponding reference material were not measured at the same beamline (Cr-, La-, and Ce-N-C) the $S_0^2$ factor multiplied by the coordination number is reported.

The metal valence state and quantification of nitrogen-moieties of the M-N-C catalysts were analyzed by XPS. The XPS was performed on a Kratos AXIS Supra spectrometer with a monochromatic Al Kα source (with an emission current of 15 mA and X-ray power of 225 W, while for the high-resolution spectra the emission current and power are 20 mA and 300 W, respectively). No charge neutralization was employed as all these samples are highly conductive, carbon-based catalysts. Survey spectra were obtained at a pass energy of 160 eV from 1400 eV to 5 eV at a step size of 1 eV. The High-resolution C 1s, N 1s and O 1s spectra were obtained at a pass energy of 20 eV with a 0.1 eV step size. The metal spectra were obtained at a pass energy of 40 eV with a 0.1 eV step size. A C 1s spectra was also obtained at a pass energy of 40 eV for calibration. CasaXPS software was used to analyze and fit all spectra and all spectra were calibrated based on the sp$^2$ carbon (284 eV). A Linear background was employed for the C 1s and N 1s spectra, while a Shirley background was used for the O 1s and metal spectra. The sp$^2$ carbon in the C 1s spectra was fit with a 80% Gaussian/20% Lorentzian function and all other spectra were fit using a 70% Gaussian/30% Lorentzian function. All fitting parameters employed in this work are based on previous fittings from our group that are reported in the literature references here for this class of M-N-C catalysts[63–66]. X-ray diffraction (XRD) patterns were collected on a Rigaku Ultima-III powder X-ray diffractometer. Inductively coupled plasma mass spectrometry (ICP-MS) on an Aligent 5110 was performed to accurately quantify the low metal content of the M-N-C catalysts. N$_2$ physisorption was recorded on a Micromeritics 3Flex Analyzer at 77 K, using a low-pressure dosing mode (5 cm$^3$ g$^{-1}$). The surface area and pore size distribution were obtained using the Brunauer–Emmett–Teller method and the non-local density functional theory model (NLDFT), respectively. Raman spectra were obtained with a 633 nm laser and 600 g.mm grating, using a Horiba LabRAM-HR.

## Preparation of the working electrode%
A catalyst ink was created by mixing 10 mg of catalyst in a solution of 680 μL IPA, 300 μL MilliQ water and 20 μL of a 5 wt% Nafion solution and sonicated for 1 h. For linear sweep voltammetry (LSV) experiments, 12.35 μL of catalyst ink was drop cast on a glassy carbon electrode (0.247 cm$^2$) for a catalyst loading of 0.5 mg cm$^{-2}$. For chronoamperometry experiments an AvCarb MGL370 carbon paper was used as the working electrode. The carbon paper was pre-treated by plasma cleaning for 15 min to remove the PTFE layer and then subsequently washed in 3 M H$_2$SO$_4$ and MilliQ water to create a hydrophilic surface. After the pretreatment, 12.5 μL of catalyst ink was drop cast on the carbon paper with a 0.25 cm$^2$ geometric working area for a catalyst loading of 0.5 mg cm$^{-2}$.

## Linear sweep voltammetry
Note that the applied potential for all electrochemical tests is not iR corrected. From PEIS measurements (from 1 MHz to 1 Hz at 7 points

per decade), the solution and contact resistances are small at 13.5 Ω, aided by the high concentration of $KNO_3$, which dissociates into $K^+$ and $NO_3^-$ ions, creating a strong electrolyte with a high conductivity. To evaluate the onset potentials for the HER, $NO_2RR$, and $NO_3RR$, LSV was performed. All LSV measurements were obtained on a Biologic potentiostat, using a rotating disk electrode in a single compartment cell. A glassy carbon electrode (0.247 cm²), graphite rod and reversible hydrogen electrode are used as the working, counter, and reference electrodes, respectively. To evaluate the HER onset potential, a 0.05 M PBS electrolyte was used. For the $NO_2RR$ onset potential LSV was performed in a 0.05 M PBS + 0.01 M $KNO_2$ electrolyte. For the $NO_3RR$ onset potential LSV was performed in a 0.05 M PBS + 0.16 M $KNO_3$ electrolyte. LSV curves were obtained by sweeping reductively from 1 V to −1 V vs. RHE at 5 mV s⁻¹ at a rotation speed of 1600 rpm. The reaction onset potential was determined to be the potential at which a current density of 0.4 mA cm⁻² was reached.

## Electrocatalytic nitrate reduction

Electrochemical nitrate reduction measurements were carried out in a customized two compartment H-cell, as shown in Fig. S66 separated by a Celgard 3401 porous polypropylene membrane (used without pre-treatment). Electrochemical measurements were recorded using an AutoLab potentiostat. A standard three-electrode system was used with a reversible hydrogen electrode (HydroFlex®) and a graphite rod as the reference and counter electrode, respectively. The electrolyte used for all nitrate reduction experiments is a 0.05 M phosphate buffer solution (PBS) and with 0.16 M $NO_3^-$ ($KNO_3$). The electrolyte pH was measured to be pH = 6.3 ± 0.05 using an Orion Dual Star pH probe. Error bars are obtained from measurements on three independent electrolytes. Prior to electrochemical tests, $N_2$ gas (research grade 99.9995% - PraxAir) is purged in both the working and counter chambers for 30 min at 80 sccm, which contain 30 mL and 25 mL of electrolyte, respectively. To assess if saturating the electrolyte with $N_2$ or Ar has an impact on the equilibrium potentials of the $NO_3RR$ processes, open circuit voltage measurements were performed under $N_2$ or Ar saturation and showed negligible differences as shown in Fig. S67. Potentiostatic tests are performed for 2 h under constant stirring at a constant $N_2$ gas flow rate of 20 sccm to the working chamber. Potentiostatic tests were performed at potentials of −0.20, −0.40, −0.60 and −0.80 V vs. RHE. After the electrolysis, the electrolyte in both the working and counter chambers was sampled and tested for $NH_3$ and $NO_2^-$. Note that at the pH = 6.3 ± 0.05 conditions utilized in this work, the pKa of ammonia has not been reached (pKa 9.2), therefore the ammonia produced is in the protonated form ammonium ($NH_4^+$). However, in the alkaline conditions present during the Berthelot method for detection, the $NH_3/NH_4^+$ is present in the $NH_3$ form. Therefore, the discussion in this manuscript is for the detected product ammonia ($NH_3$). Note our previous work has demonstrated that these atomically dispersed M-N-C catalysts have been shown to be inactive for the reduction of $N_2$ under these conditions[24].

NO3RR tests in alkaline media (pH = 13.8 ± 0.17) were performed using a 1 M KOH + 0.16 M $KNO_3$ electrolyte. Due to the extremely high $NH_3$ yield rate, potentiostatic tests were performed for 15 min at potentials of −0.20, −0.40, −0.60, and −0.80 V vs. RHE.

## Electrocatalytic nitrite reduction

Electrochemical nitrite reduction experiments were performed analogously to nitrate reduction experiments. The electrolyte is 0.05 M PBS and 0.01 M $NO_2^-$ ($KNO_2$) (pH = 6.26 ± 0.11). Due to the higher $NH_3$ yield rates of the more active $NO_2RR$, potentiostatic tests were performed for 30 min. Potentiostatic tests were performed at potentials of −0.20, −0.40, −0.60, and −0.80 V vs. RHE. After each electrolysis, the electrolyte in the working chamber was sampled and tested for $NH_3$ detection.

## Isotope-labeling experiments

**Isotopic nitrate $^{15}NO_3^-$ labeling.** Isotopic $^{15}NO_3^-$ labeling experiments were performed by using $K^{15}NO_3$ (99% - Cambridge Isotopes) as the isotopic nitrate source with a 0.05 M PBS + 0.16 M $K^{15}NO_3$ electrolyte (pH 6.3 ± 0.05). Isotopic $^{15}NO_3^-$ experiments were performed using a metal free N-C catalyst. Using a metal free N-C catalyst simultaneously allows the source of the N in the produced $NH_3$ to be confirmed, while also demonstrating that the even the metal free N-moieties in the N-C catalysts can catalyze the $NO_3RR$ (and is not from N-originating from catalyst decomposition).Electrolysis was performed at −0.40 V for 4 h, after which the electrolyte in the working chamber was samples and the produced $^{15}NH_3$ was quantified by $^1H$ NMR.

**Isotopic nitrite $^{15}NO_2^-$ labeling.** Isotopic $^{15}NO_2^-$ labeling experiments were performed by using $Na^{15}NO_3$ (98%⁺ - Cambridge Isotopes) as the isotopic nitrite source. Isotopically labeled $^{15}NO_2^-$ at concentrations of 100, 10, and 1 ppm were doped into the 0.05 M PBS + 0.16 M $KNO_3$ (10,000 ppm $NO_3^-$) electrolyte. Potentiostatic tests were performed at −0.40 V vs. RHE for 6 h, under constant stirring and a $N_2$ gas flow of 20 sccm to the working chamber. The electrolyte was sampled at 2-, 4-, and 6-h time intervals and the $^{15}NH_3$ was quantified by $^1H$ NMR.

**Calculation of the yield and faradaic efficiency.** Note that all error bars in this work are determined from triplicate experiments on independently prepared electrodes and are then calculated using a 90% confidence interval.

For the nitrate ($NO_3^-$) and nitrite ($NO_2^-$) reduction reactions, the $NH_3$ yield rate was calculated by Eq. (1).

$$Yield_{NH_3} = \frac{c_{NH_3} * V}{Mw_{NH_3} * t * A_{electrode}} \qquad (1)$$

For the nitrate ($NO_3^-$) and nitrite ($NO_2^-$) reduction reactions, the $NH_3$ Faradaic efficiency was calculated by Eq. (2).

$$FE_{NH_3} = \frac{n * F * c_{NH_3} * V}{Mw_{NH_3} * Q} \qquad (2)$$

In the $NO_3RR$, the nitrate ($NO_3^-$) to nitrite ($NO_2^-$) reduction Faradaic efficiency was calculated by Eq. (3).

$$FE_{NO_2^-} = \frac{n * F * c_{NO_2^-} * V}{Mw_{NO_2^-} * Q} \qquad (3)$$

where $c_{NH_3}$ is the concentration of $NH_{3(aq)}$ (μg mL⁻¹), $V$ is the volume of the electrolyte (mL), $Mw_{NH_3}$ is the molar mass of $NH_3$ (17.031 g mol⁻¹), $t$ is the duration of the chronoamperometric measurement (h), the surface area of the working electrode, $A_{electrode}$ is 0.45 cm². The number of electrons transferred, $n$, for $NO_3^-$ to $NO_2^-$ is $n = 2$ and for $NO_3^-$ to $NH_3$ is $n = 8$.

$F$ is Faraday's constant (96,485 C mol⁻¹), the concentration of $NO_2^-$ (μg mL⁻¹), $C_{NO2-}$, where the molar mass of $NO_2^-$, $Mw_{NO2-}$ is (46.005 g mol⁻¹) and the charge during the chronoamperometric measurement, Q.

**Product detection.** For typical, non-isotopically labeled $NO_3RR$ and $NO_2RR$ electrolysis, products were quantified using an ultraviolet-visible (UV−Vis) spectrophotometer (Shimadzu, UV−2600).

**Determination of ammonia.** Ammonia was quantified by UV−Vis using the Berthelot reaction. In cases producing large concentrations of $NH_3$, the working electrolyte solution was diluted such that the absorbance would fall within the range of the calibration curve. Specifically, 2 mL of the electrolyte (or diluted electrolyte if needed) was pipetted into a vial. To this, 2 mL of a 1 M NaOH solution that contains 5 wt% salicylic acid, and 5 wt% sodium citrate was added. Next, 1 mL of a 0.05 M

NaClO and 0.2 mL of a 1 wt% $C_5FeN_6Na_2O$ (sodium nitroferricyanide) was added to the solution. The solution was incubated in the dark at ambient condition for 1 h, then UV-Vis spectra were recorded. The concentration of $NH_3$ is determined using the maximum absorbance at a 655 nm wavelength, when compared to the generated calibration curves, Figs. S68–69.

**Determination of nitrite.** Nitrite was quantified by UV–Vis using a commercial $NO_2^-$ assay kit (Spectroquant), based on the Griess test. In cases producing large concentrations of $NO_2^-$, the working electrolyte solution was diluted such that the absorbance would fall within the range of the calibration curve. Specifically, 2 mL of the electrolyte (or diluted electrolyte if needed) was placed in a vial. To this, 22 mg of the assay reagent was added to the electrolyte and incubated in the dark at ambient conditions for 10 min. After incubation, the UV–Vis spectra were taken, and the concentration of nitrite was determined at a maximum absorbance of 540 nm wavelength, when compared to the generated calibration curves, Fig. S70.

**NMR determination of ammonia.** For isotopic labeling experiments, nuclear magnetic resonance (NMR) spectroscopy was used to detect and quantify both $^{14}NH_3$ and $^{15}NH_3$. Dimethylsulfoxide-d6 (DMSO) and 3-(trimethylsilyl)-1-propanesulfonic acid sodium salt (DSS) were used as the locking solvent and internal standard, respectively. For the NMR test solution, 580 μL of the electrolysis electrolyte, 25 μL of DMSO, 20 μL of 3 M $H_2SO_4$, and 75 μL of 6 mM DSS (made with Millipore water) are mixed. The NMR spectrum was obtained on a Bruker CRYO 500 MHz spectrometer. The signal from $H_2O$ was restrained for better accuracy by applying the solvent suppression method during acquisition. Topspin 4.0.8 software was used to process the NMR data. Standard NMR calibration curves for standard $^{14}NH_3$ and isotopically labeled $^{15}NH_3$ are shown in Fig. S71.

**Computational details.** All the DFT calculations were performed with the generalized gradient approximation approach and projector augmented-wave pseudopotentials[67,68] using the Vienna Ab initio Simulation Package[69–71]. To account for the van der Waals interactions, an optB86b-vdW functional was used[72–75] with a gamma centered $8 \times 8 \times 1$ (M-$N_4$ sites, pyridinic N, graphitic N) or $3 \times 3 \times 1$ k-mesh (pyrrolic NH) and Fermi-smearing. Sigma and the plane-wave basis cutoff was set to 0.03 and 400 eV, respectively. We approached modeling of the M-$N_4$ sites in the M-N-C catalysts by creating two carbon atom vacancies in the graphene, in which we place a metal atom[27,65,76–78]. Structures were then optimized as neutral, using unrestricted open shell spin density and with careful evaluation of the effect the initial magnetic moment has on the self-consistent solution of the electronic problem[79,80]. Adsorption energies are given relative to the M-$N_4$ defect with a total magnetic moment corresponding to the lowest electronic energy. M-$N_4$, pyridinic N, and graphitic N sites were modeled using $4 \times 4$ orthorhombic single-layer graphene cell with the dimensions of $9.84 \times 8.52$ Å. Pyrrolic NH defect was modeled using a hexagonal single layer $17.04 \times 17.04$ Å unit cell. Structures of all the unit cells used for the DFT calculations are shown on Fig. S72. A vacuum region of 20 Å was applied in all cases. All structures were optimized by allowing all atoms to relax (including adsorbents) while the cell parameters were kept fixed at the DFT optimized value for graphene. The criteria for the convergence of the electronic energy were set to $1 \times 10^{-5}$ eV. The forces were converged to 0.01 eV Å$^{-1}$. All calculations were performed as spin polarized calculations. The Pearson correlation coefficient was calculated using the python function np.corrcoef, which utilizes the equation below.

$$R = \frac{\sum (x_i - \bar{x}) * (y_i - \bar{y})}{\sqrt{\sum (x_i - \bar{x})^2 * \sum (y_i - \bar{y})^2}}$$

Where R, is the correlation coefficient, $x_i$ and $y_i$ are the x and y values in samples, and $\bar{x}$ and $\bar{y}$ are mean values for the x and y variables. The computational approach for the calculation of Gibbs free energy of reactions and considered steps in the $NO_3RR$ and $NO_2RR$ mechanism are explained in details of our recent work[24]. However, it must be pointed out that all the Gibbs free energies were calculated at an experimentally determined pH of 6.3. We also emphasize that M-N-C catalysts are challenging to simulate, and the reliability of the computational results depends on many factors (e.g., see refs. 81,82 and SI note 4). Although one should pay attention to the accuracy of the absolute numbers, relative values and trends are more reliable.

## Data availability

The computational data for Figures and Supplementary Figures is available in X repository. All other relevant data supporting for the findings in this work are available from the authors upon request.

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

## Acknowledgements

This work was supported in part through collaboration with Sandia National Laboratories (SNL), funded by the U.S. Department of Energy's Office of Energy Efficiency and Renewable Energy (EERE), under the Advanced Manufacturing Office (AMO), FOA Number DE-LC 000L059. Sandia National Laboratories is a multi-mission laboratory managed and operated by National Technology & Engineering Solutions of Sandia, LLC, (NTESS) a wholly owned subsidiary of Honeywell International, Inc., for the U.S. DOE's National Nuclear Security Administration under contract DE-NA-0003525. The views expressed in the article do not necessarily represent the views of the U.S. DOE or the United States Government. This paper describes objective technical results and analysis. Any subjective views or opinions that might be expressed in the paper do not necessarily represent the views of the U.S. Department of Energy of the United States Government. This article has been authored by an employee of NTESS, and the employee owns all right, title and interest in and to the article and is solely responsible for its contents. The United States Government retains and the publisher, by accepting the article for publication, acknowledges that the United States Government retains a non-exclusive, paid-up, irrevocable, world-wide license to publish and reproduce the published form of this article or allow others to do so, for United States Government purposes. The DOE will provide public access to these results of federally sponsored research in accordance with the DOE Public Access Plan https://www.energy.gov/downloads/doe-public-access-plan.

The authors acknowledge the use of facilities and instrumentation at the UC Irvine Materials Research Institute (IMRI), which is supported in part by the National Science Foundation through the UC Irvine Materials Research Science and Engineering Center (DMR-2011967). XPS was performed at the UC Irvine Materials Research Institute (IMRI) using instrumentation funded in part by the National Science Foundation Major Research Instrumentation Program under grant no. CHE–1338173. IM would like to thank UNM Center for Advanced Research Computing, supported in part by the National Science Foundation, for providing the high-performance computing resources used in this work. This research also used resources of the National Energy Research Scientific Computing Center (NERSC), a U.S. Department of Energy Office of Science User Facility located at Lawrence Berkeley National Laboratory, operated under Contract No. DE-AC02-05CH11231. The work from the Fritz Haber Institute team has been supported by the European Research Council (ERC-725915, OPERANDOCAT).

This research used resources of the Advanced Photon Source, a U.S. Department of Energy (DOE) Office of Science User Facility operated for the U.S. DOE Office of Science by Argonne National Laboratory under contract No. DE-AC02-06CH11357.

The authors would like to sincerely thank the beamline scientists Andrea Zitolo and Emiliano Fonda (SOLEIL), Giuliana Aquilanti (ELETTRA), as well as Michael Haumann and Götz Schuck (BESSY II) for their help in obtaining synchrotron measurements. Raman was conducted in UC Irvine's Horiba Institute for Mobility and Connectivity[2] (HIMAC[2]).

## Author contributions

E.M. and Y.L. designed the experiments, analyzed the data, and wrote the original draft of the paper. E.M. synthesized the catalysts, performed catalyst characterization, and performed the electrochemical experiments. I.M. conducted the first-principles calculations and contributed to writing the manuscript. M.R., A.M., J.T., and B.C. performed XAS experiments and data processing. W.Z., X.Y., and X.P. performed EELS experiments. Y.L., Y.H., A.L., S.G., performed catalyst characterizations. I.Z., E.S. and P.A. supervised the project. P. A, E.M., and Y.L. conceived the project. All authors contributed to the preparation of the manuscript.

## Competing interests

The authors declare no competing interests.
