## [Peer Review File · Nature Communications]

REVIEWER COMMENTS

Reviewer #1 (Remarks to the Author):

In this work, Atanassov et al. synthesized a set of atomically dispersed M-N-C catalysts and applied them in the electrochemical reduction of NO_3^- and NO_2^- under neutral media. The authors have presented comprehensive data and concluded that the mechanisms for the reduction of NO_3^- are varied in the direct $8e^-$ path or the cascade $2e^-+6e^-$ path for different M-N-C catalysts. However, some descriptions and result explanations are not satisfactory. I suggest that this paper should be revised significantly and then submitted to other more special journals.

Major points:

1. It was unusual to correlate the reduction activity of nitrite or reductive adsorption of nitrate with the selectivity of NH_3 . In principle, the selectivity of NH_3 may not be properly discussed without screening all the byproducts.
2. The results demonstrated in Figure 3e and Figure 4c seem paradoxical. The NH_3 produced from 14NO_3^- was barely affected by the 15NO_2^- and the concentration of 15NH_3 in the products was far higher than the concentration of 15NO_2^- in the electrolyte (Figure 4c), implying that the reduction of NO_2^- should be favored even in low concentrations. However, large part of NO_2^- was still observed (Figure 3e). In addition to the Fe-N-C, the $15\text{NO}_2^- + 14\text{NO}_3^-$ experiment should be conducted for other M-N-C catalysts.
3. The XPS results indicated the multiple composition of the metal species are contained in most of the M-N-C catalysts, making the M-N₄ structure proposed as the active center ambiguously.
4. Experimentally, different atomically dispersed metal have different valence state(s). For example, Fe should be in the +2/+3 oxidation state, Cr has a mixed oxidation state of +3 and +4, while Mo shows a oxidation state between +4 and +6. Did the theoretical models appropriately describe the different oxidation states of different metal centers?
5. Most linear correlations were rather poor when considering all elements.

Minor points:

6. The XAS characterization for other M-N-C should be provided.
7. XAS k-space fitting result should be present.
8. WT-EXAFS should be compared with the standards.
9. The evaluated M-N coordination number in Table S3 was far higher than M-N₄.

10. Peak shifting can be observed in the XRD patterns (Figure S4).
11. The XPS of Fe-N-C is present in both Figure S10 and Figure S11, the quality of the data was very low as compared to other M-N-C with similar metal loading.
12. Line 152, "Figure 2d-e." should be "Figure 2d-f"?
13. Line 232 "NO₃- " should be "NO₂-"?
14. The title of Table S5 should be the results of M-N-C rather than metal free samples?

Reviewer #2 (Remarks to the Author):

The author prepared 13 kinds of single atomic catalysts with different elements, and applied them to the nitrate electroreduction. Through experimental analysis and theoretical calculation, the author proposed descriptors for evaluating the activity of M-N-C catalysts for the NO₃RR and NO₂RR. Unfortunately, there are many problems, and there is a lack of detailed explanation for supporting key conclusions. Therefore, I do not recommend publishing this paper at its present version. A revised might be reconsidered if the authors can address all the following concerns.

1. For single atomic catalysts, XAS test and fitting analysis are necessary, otherwise the results are not convincing. Although the HRTEM images can intuitively display the morphology of single atoms, there is a great contingency. XRD, XPS and EDS can not confirm the single atomic structure.
2. The author thinks that the single atom structure should be M-N₄ (Fig. 1b). why do elements exhibit multiple valence states?
3. For fig. 3b, why should HER be taken as the baseline and deducted? The analysis of "LSV with HER subtracted" for NO₃RR and NO₂RR is unreasonable. This treatment means that the authors believe that HER and NO₃RR processes coexist at a specific potential. However, the subsequent test results show that for some elements, HER almost does not occur or can be ignored. Obviously, this is very contradictory to the author's operation. Actually, in the solution where H⁺/H₃O⁺ and NO₃⁻ coexist, NO₃RR may be superior to adsorption or subsequent hydrogenation process. Thus, directly deducting HER is unreasonable.
4. All the data provided in Supplementary Information should be reflected in the manuscript.
5. If the FE of HER need to be discussed, the author should use gas chromatography for quantitative analysis. Whether the photos of bubble overflow is not valid evidences.

6. The author should discuss in detail why Yield $\text{NO}_2\text{RR-NH}_3$ and FEN $\text{O}_3\text{RR-NH}_3$ are linear? What does a linear relation mean? The author lacks the necessary derivation process, What is the core of this work?

7. Necessary explanations should be provided, such as “[$\text{NO}_2\text{RR YR NH}_3$] | [$\text{NO}_2\text{RR FE NH}_3$] vs. $\text{NO}_3\text{RR FENH}_3$ ”. Meanwhile, the author should explain why to do such complex operations, or just find a relationship?

8. For all single atomic catalysts provided by the author, the current density is very small (the maximum current density is less than 20 mA cm^{-2}). At present, the current density and FE of NO_3RR to ammonia are very high (FE $>90\%$; current density up to 2 A cm^{-2}). What is the significance or advantage of monatomic catalyst?

9. According to Marcus-type theory (MPIP. J. Club-Mainz., 2008, 1, 1–13; Chem. Sci., 2013, 4, 2710–2723), the transfer of electrons occurs one by one. Why do you consider the first two steps together in the DFT calculations?

10. For the mechanism of [$\text{*+NO}_3 \rightarrow \text{*NO}_3$], one electron is involved. For the mechanism of [$\text{*+NO}_3 \rightarrow \text{*NO}_2$], one electron and two protons are involved. When the above mechanisms are beyond the CHE model, how did you calculate these decoupled electron and proton energies?

11. For the oxidative associative adsorption mechanism, [$\text{*+NO}_3 \rightarrow \text{*NO}_3$], if the electron energy is $-eU$, the Gibbs free energy of this step increases along with decreasing potential. It is well known that the negative potential corresponds to better performance in the reduction reaction. How do you consider this question?

12. Line 308, you have lost a minus symbol ($\Delta rG \sim -4 \text{ eV}$). It is very troublesome in rigorous scientific research.

13. In the manuscript figure 5(a), the author said that the Mo-N $_4$, La-N $_4$, Ce-N $_4$ and W-N $_4$ sites could strongly adsorb NO_3^- in a dissociative manner, forming an *O surface species and a free NO_2^- molecule. However, the La-N $_4$ and Ce-N $_4$ perform the competition of *NO_2 and *NO_3 . Meanwhile, the author said that the Cr-N $_4$, Mn-N $_4$, Fe-N $_4$, and Ru-N $_4$ sites, associative and dissociative adsorption of NO_3^- is likely a competitive process that also depends on the cell potential. However, the Cr-N $_4$ prefers to form an *O surface species and a free NO_2^- molecule. In addition, the conclusion that Rh-N $_4$ has poor NO_3RR activity from the DFT descriptors is not rigorous.

14. The author calculated the Gibbs free energy at experimental pH of 6.3, but the experimental condition is pH=14. The influence of pH may need to be considered to deepen the mechanism, as the nitrite/nitrate reduction showed pH dependence (Chem. Commun., 2014, 50, 2148-2151; J. Am. Chem. Soc., 2018, 140, 2012-2015).

15. The solution effect is very important in electrochemistry, but you didn't use any solution correction in the DFT calculations. In addition, the spin effect was not taken into account.

Reviewer #3 (Remarks to the Author):

The work by Murphy et al. is an extensive consideration of M-N-C electrocatalysts for NO₃RR, targeting NH₄⁺ in neutral media. Considering a wide range of TM centers, the authors thoughtfully probe the role of the NO₂⁻ intermediate and a potential cascade reaction through an intriguing isotopic labelling study, with complementary consideration of different modes of adsorption of NO₃⁻ and NO₂⁻ by DFT. The topic is timely and of interest to the readers of Nature Communications. Although the work is thorough and compelling, there are a couple points which must be addressed before publication:

Introduction: Page 2 line 62, "Previous attempts..." It seems remiss to not put this work in context with recent work considering competitive adsorption of H and NO₃ on the surface of 3d and 4d transition metals in neutral media and implications regarding NO₃RR FE and selectivity to NH₄⁺ (<https://doi.org/10.1021/jacs.2c05673>).

Figures 2 j-l, S10-14, and discussion of XPS spectra: In general, the author's description of XPS data collection (incident X-ray source and energy, use of charging neutralization), binding energy scale calibration, sufficiently wide binding energy window collected to observe background regions and corresponding background selections, and fitting procedures and justifications are lacking in both literature support and discussion. This could be provided in the SI.

Pg. 9 line 202 & Figure 3b "HER subtracted LSV..." Subtracting HER current as a "background" current implies that the rate of HER is unchanged by the addition of NO₃⁻ (presumably leading to competitive adsorption between H and NO₃⁻ on the surface, lacking discussion in the present work), which does not seem supported by the high NO₃RR FE generally observed. Please present the LSV without this subtraction, overlaying HER if a comparison is desired.

Fig 3 and methods: FE and selectivity data are based upon CA holds for a fixed amount of time (2h), both across samples and applied potentials. However, the authors (and others in literature) clearly articulate that NO₂⁻ is in many cases an intermediate, whose abundance in a batch cell will inherently depend upon the extent of conversion, which is uncontrolled via electrolysis time. The authors should present the extent of conversion of NO₃⁻ for different catalysts and the applied potentials considered, and if these differ greatly comment on the implications in conclusions made.

Fig 4b line 285 and pg 15 line 359 state that Ru and Rh are excluded from fitted correlations because of gaseous products (H₂ based on verbiage in S37), and Co as well (perhaps because of gaseous products pre line 359, but not detailed in the figure caption). The implications of this exclusion should be discussed in more detail. It seems disingenuous to call these "outliers" excluded from fits correlating interesting ties between NO₂⁻ and NO₃⁻, but then in the conclusion (lines 389-391) claim "a universal contribution from the NO₂⁻ intermediate" and that DFT calculations of NO₃⁻ and NO₂⁻ adsorption

“explain the selectivity for each metal center.” The authors should be more clear about how far their conclusions go, their limitations (perhaps validity only in cases with near 100% FE to NO₃RR?), and the assumptions that go into them.

Pg 21 line 519: although the authors have previously shown that M-N-C catalysts are not active to reduce N₂, does the presence of N₂ shift equilibrium potentials (and resultant current densities or product distributions) of NO₃⁻ reduction processes in comparison to e.g. saturating with Ar?

And a few general comments:

- At the neutral conditions here the species formed from NO₃⁻ reduction should be NH₄⁺ (ammonium) as pH 7 is below the pK_a of ammonia. While the detection method of the Berthelot reaction is robust for its detection, the nomenclature is important due to the implications on proton demand and the product remaining dissolved in solution.

- Error bars are throughout the main text, but the source of them is not clearly identified in the manuscript or SI.

- - What is the justification for the authors using geometric surface area for current density normalization and ammonia/ammonium yield calculations? How has solution and contact resistance been accounted for? When performing two-cell selectivity measurements, are solution and contact resistances accounted for during the measurement? If not, what are the implications on the chronoamperometric potentials measured?

Reviewer #1

In this work, Atanassov et al. synthesized a set of atomically dispersed M-N-C catalysts and applied them in the electrochemical reduction of NO_3^- and NO_2^- under neutral media. The authors have presented comprehensive data and concluded that the mechanisms for the reduction of NO_3^- are varied in the direct $8e^-$ path or the cascade $2e^-+6e^-$ path for different M-N-C catalysts. However, some descriptions and result explanations are not satisfactory. I suggest that this paper should be revised significantly and then submitted to other more special journals.

Major points:

1. It was unusual to correlate the reduction activity of nitrite or reductive adsorption of nitrate with the selectivity of NH_3 . In principle, the selectivity of NH_3 may not be properly discussed without screening all the byproducts.

We thank the reviewer for raising the discussion of this key point. These correlations may seem “unusual”, but they are exactly the novelty of this work and were strongly supported by our experimental evidence.

To a certain extent, the “usual” thing is that the current NO_3RR literature consists of papers studying single catalysts with DFT simulations of one or several pre-defined full reaction coordinates. The correlation between the observed activity and lowest thermodynamic step is indeed “popular” but is scientifically meaningless. As suggested by Jens Norskov and other computational catalysis authorities, DFT result only make sense when it is reflecting a trend rather than a single active site or reaction energy.

- a) In this work, we experimentally showed the ease of reducing NO_2^- to NH_3 (Fig.4d) on diverse active sites, as compared to the reduction of NO_3^- to NH_3 . This demonstrates that in the NO_3RR , the sequential $6e^-$ transfer is not the rate limiting step (as suggested by several studies in the literature), rationalizing the focus on the first $2e^-$ transfer (NO_3^- to NO_2^- or $^*\text{NOOH}$). Additionally, we have experimentally shown strong evidence of the complex role of the NO_2^- intermediate in the NO_3RR (Fig. 4a-c and new Fig. S44). Therefore, the computational study of the steps before NO_2^- (e.g., reductive adsorption of nitrate) and their correlation to NH_3 selectivity have a solid experimental base and the trends reflected by this correlation have profound scientific significance.

- b) For those “usual” papers in the literature, the NO₃RR process was simply assumed as a direct 8e⁻ process and the observed NO₂⁻ was usually assumed as a byproduct instead of a highly active reaction intermediate. However, our isotopic analysis on diverse metal centers (Fig. 4a-c and new Fig. S44) demonstrates the strong activity for the NO₂⁻ reduction to NH₃ in the presence of highly concentrated NO₃⁻. In other words, even for the NO₃RR case of 100% F_{ENH₃} with zero observed NO₂⁻ in the bulk, one still cannot confirm the direct 8e⁻ transfer pathway, since there could be the 2e⁻+6e⁻ scenario that the NO₂⁻ → NH₃ was faster than NO₃⁻ → NO₂⁻. Therefore, we proposed a prototype that the consumption/reduction rate of nitrite (NO₂⁻) could largely impact the NH₃ selectivity under a 2e⁻+6e⁻ framework. This is the original “unusual” novelty of this work, based on solid experimental support and a reasonable scientific hypothesis.
- c) We totally agree with the reviewer that screening all byproducts could contribute to the discussion of NH₃ selectivity. However, under the context of this work, we have shown that NO₂⁻ and NH₃ are the two dominant NO₃RR products and the selectivity of these two adds up close to 100% for most data points. Therefore, focusing on NO₂⁻ and NH₃ will not impact the main conclusion of this work.

2. The results demonstrated in Figure 3e and Figure 4c seem paradoxical. The NH₃ produced from ¹⁴NO₃⁻ was barely affected by the ¹⁵NO₂⁻ and the concentration of ¹⁵NH₃ in the products was far higher than the concentration of ¹⁵NO₂⁻ in the electrolyte (Figure 4c), implying that the reduction of NO₂⁻ should be favored even in low concentrations. However, large part of NO₂⁻ was still observed (Figure 3e). In addition to the Fe-N-C, the ¹⁵NO₂⁻ + ¹⁴NO₃⁻ experiment should be conducted for other M-N-C catalysts.

With respect to Figure 4c, the concentration of ¹⁵NH₃ was always far lower than the concentration of ¹⁵NO₂⁻ in the electrolyte. The concentration of ¹⁵NH₃ was either less than 2 ppm (with a 10-ppm feed of ¹⁵NO₂⁻) or less than 10 ppm with (with a 100-ppm feed of ¹⁵NO₂⁻). However, the reviewer is correct in that the implication is that the reduction of NO₂⁻ is favored even in with a low NO₂⁻ concentration (in highly concentrated 10,000 ppm NO₃⁻). The reviewer correctly notes that the concentration of produced ¹⁴NH₃ does not appear to be impacted by the concentration of isotopically doped ¹⁵NO₂⁻ (either 10 or 100-ppm). This is due to impart because there is no change in the driving force for ¹⁴NO₃⁻ between either experiment as this concentration remains constant and additionally, the 6e⁻ transfer reaction from NO₂⁻ to NH₃ is significantly faster than the rate limiting step 2e⁻ transfer step from NO₃ to NO₂ and evidenced when comparing the NO₃RR and NO₂RR results. It is likely that the ¹⁵NO₂⁻ reactant is not occupying the active site for long, relative to ¹⁴NO₃⁻ reactant and therefore at these

relatively low $^{15}\text{NO}_2^-$ concentration we do not see an obvious impact on the $^{14}\text{NO}_3^-$ to $^{14}\text{NH}_3$ performance.

We thank the reviewer for suggesting this helpful tip to make the isotopic doping experiment clearer and have performed this experiment for the remaining 12 other M-N-C catalysts as shown in Figure S44 and shown below. From this figure a few interesting points are observed. First, over each M-N-C catalyst, and even metal free N-C (Figure S43), we see that with a $^{15}\text{NO}_2^-$ concentration of 10 ppm (1000x less than the 10,000 ppm $^{14}\text{NO}_3^-$), the NO_2^- can easily adsorb (or re-adsorb in the case of standard NO_3RR reactions) and reduce to $^{15}\text{NH}_3$ even at very low concentrations compared to its $^{14}\text{NO}_3^-$ reactant counterpart. Showing the ease of the NO_2RR and supporting the production consumption pathway of NO_2^- in the NO_3RR . Additionally, for 2 metals, Co-N-C and Ni-N-C, unique behavior is observed in that more $^{15}\text{NH}_3$ is produced than $^{14}\text{NH}_3$ even at a 1000:1 concentration of $^{14}\text{NO}_3^- : ^{15}\text{NO}_2^-$, highlighting the very limited activity towards the activation of NO_3^- , and high selectivity to the reduction of NO_2^- . Which is in strong agreement with the demonstrated NO_3RR and NO_2RR results shown in Figure 3e and Figure 4d of the main text.

Figure S 44. Competing NO_2RR in concentrated NO_3^- . NO_3RR electrolysis with isotopically labeled $^{15}\text{NO}_2^-$ in concentrated $^{14}\text{NO}_3^-$ for all M-N-C catalysts. All electrolysis is performed at -0.40 V vs. RHE for 2 hours. The electrolyte consists of 0.16M K^{14}NO_3 and 10 ppm of isotopically labeled nitrite, K^{15}NO_2 . For all catalysts (except Co & Ni) there is a dominating triplet indicative of the reduction of the K^{14}NO_3 to standard $^{14}\text{NH}_3$, with a smaller doublet peak for isotopic $^{15}\text{NH}_3$. For Co and Ni, very unique behavior is observed in that these atomically dispersed metal centers show poor activity for the NO_3RR , but are highly active for the reduction of NO_2^- , as shown by the dominating doublet for $^{15}\text{NH}_3$.

3. The XPS results indicated the multiple composition of the metal species are contained in most of the M-N-C catalysts, making the M-N₄ structure proposed as the active center ambiguously.

Thank you to the reviewer for bringing up this important point. This is a point of great contention in the atomically dispersed M-N-C catalyst field. It is known that there is a plurality of M-N_xC_y (Fe-N₄, Fe-N₂₊₂, N-Fe-N₂₊₂) sites present in these catalyst materials and is incorrect to declare that one has exclusively M-N₄ sites (unless one is working

with porphyrins and phthalocyanines etc.). We have been careful and explicit in the text to declare that we have confirmed through EELS, XPS and synchrotron EXAFS, the presence of $M-N_x$ coordinated sites. We have used DFT to also consider the variety of active N-moieties (pyrrolic, pyridinic and graphitic) present in these materials towards the NO_3RR (Figure S61 and Table S7). The choice to designate the $M-N_4$ site as the active site for our computational descriptors is based not only on the spectroscopic results in the current work but is also supported by prior work, establishing the $M-N-C$ catalysts synthesized via the sacrificial support method (SSM) contain $M-N_4$ sites (among other atomically dispersed $M-N_xC_y$ sites mention above) as shown through Mossbauer spectroscopy.¹

1. [Primbs, Mathias, et al. "Establishing reactivity descriptors for platinum group metal (PGM)-free Fe–N–C catalysts for PEM fuel cells." *Energy & Environmental Science* 13.8 (2020): 2480-2500.]

4. Experimentally, different atomically dispersed metal have different valence state(s). For example, Fe should be in the +2/+3 oxidation state, Cr has a mixed oxidation state of +3 and +4, while Mo shows a oxidation state between +4 and +6. Did the theoretical models appropriately describe the different oxidation states of different metal centers?

Reviewer is right that different atomically dispersed metals have different valence state(s) and to account for that, all the calculations have been done as spin polarized allowing the convergence to the energetically lowest electronic state *of the periodic structure*. We find that Fe-containing complex $Fe-N_4$ has 2 unpaired electrons (1.98 from VASP output), corresponding to Fe +2 state, $Cr-N_4$ has 3.92 unpaired electrons corresponding to +2 state, and $Mo-N_4$ has 2.7 unpaired electrons corresponding to +3/+4 state. Please note that the electron density on metal atoms is strongly dependent on the metal's environment (type of atoms in the metal atom's coordination shell as well as the electron density distribution in the periodic structure i.e., distribution of metal states around the Fermi level).

5. Most linear correlations were rather poor when considering all elements.

The main linear correlations between computational and experimental activity descriptors for NO_3RR and NO_2RR have r values of 0.73 and 0.72, respectively. Given the plurality of active sites in atomically dispersed catalysts, complex nature of the multi-step proton-electron transfer reactions and pairing of a single computational & experimental descriptor, achieving these r values is quite good. By definition, a strong correlation is $r > 0.70 - 0.75$ (depending on the source), therefore, the resulting r values obtained here constitute a strong (or moderate-strong) correlation. Certain elements such as Ru & Rh are considered as outliers due to their poor NO_3RR performance and presence of gas phase products. Similarly, the Co metal center has shown very unique

activity demonstrating poor activation of NO_3^- , while excellent activation of NO_2^- (as shown through our isotopic doping experiments over Co-N-C in Figure S44).

Minor points:

6. The XAS characterization for other M-N-C should be provided.

Additional synchrotron based XAS experiments were performed for additional M-N-C catalysts and corresponding references and has been added in Figure S 10 -12. With a full discussion on the XANES and EXAFS analysis in Supplementary notes 1 & 3, respectively.

Figure S 10. XANES, FT-EXAFS and EXAFS spectra for the 3d metals (a-c) Cr-K edge in Cr-N-C, (d-f) Mn-K edge in Mn-N-C, (g-i) Fe-K edge in Fe-N-C and (j-l) Co-K edge in Co-N-C. See Supplementary note 1 for a full discussion on the XANES interpretation and determination of metal oxidation states and Supplementary Table 2 and

Supplementary note 3 for a full discussion on the EXAFS analysis.

Figure S 11. XANES, FT-EXAFS and EXAFS spectra 3d metals (a-c) Ni-K edge in Ni-N-C and (d-f) Cu-K edge in Cu-N-C and XANES spectra for 4d metals (g) Mo-K edge in Mo-N-C and (h) Pd-K edge in Pd-N-C. See Supplementary note 1 for a full discussion on the XANES interpretation and determination of metal oxidation states and Supplementary Table 2 and Supplementary note 3 for a full discussion on the EXAFS analysis.

Figure S 12. XANES, FT-EXAFS and EXAFS spectra for 4d and f-metals (a-c) Rh-K edge in Rh-N-C, (d-f) La-L edge in La-N-C and (g-i) Ce-L edge in Ce-N-C. See

Supplementary note 1 for a full discussion on the XANES interpretation and determination of metal oxidation states and Supplementary Table 2 and Supplementary note 3 for a full discussion on the EXAFS analysis.

7. XAS k-space fitting result should be present.

The k-space analysis has been added in addition to the XANES and EXAFS spectra in Figure S10 – S12, see above. All EXAFS fitting parameters have been provided in Table S2 and S8. With a complete discussion on the EXAFS fittings in Supplementary note 3.

Table S2. Structure parameters from the EXAFS fittings for the M-N-C catalysts including coordination number, interatomic distances (Metal-Oxygen, RM-O and Metal-Nitrogen, RM-N) and disorder factor σ^2 . Uncertainties are given in the parenthesis.

Sample	S_0^2 factor	Coordination number	$R_{M-N/M-O}$ (Å)	$\sigma^2_{M-N/M-O}$ (Å ²)	ΔE_0 (eV)	R factor
Cr-N-C		2.4 ± 0.3	1.97 ± 0.001	0 ± 0.003	5.31 ± 2.8	0.81%
Cr ₂ O ₃	0.56	6	1.96 ± 0.01	0.002 ± 0.002	4.18 ± 1.1	0.13%
Mn-N-C		7.9 ± 1.4	2.12 ± 0.1	0.017 ± 0.001	-5.52 ± 0.7	0.01%
MnO	0.48	6	2.2 ± 0.03	0.002 ± 0.003	-5.08 ± 2.1	0.46%
Fe-N-C		5.98 ± 1.1	2.12 ± 0.1	0.009 ± 0.005	8.49 ± 1.6	0.37%
Fe ₃ O ₄	0.53	5.3	2.03 ± 0.1	0.005 ± 0.005	8.56 ± 1.3	0.25%
Co-N-C		8.4 ± 0.9	1.92 ± 0.02	0.021 ± 0.002	-5.82 ± 1.1	0.46%
Co ₃ O ₄	0.54	5.3	1.91 ± 0.02	0 ± 0.005	-6.41 ± 3.4	0.90%
Ni-N-C		1.8 ± 0.3	1.89 ± 0.2	0.01 ± 0.003	-7.58 ± 1.7	0.21%
Ni(OH) ₂	0.99	6	2.08 ± 0.06	0.006 ± 0.004	-11.41 ± 2.9	0.60%
Cu-N-C		5.2 ± 0.4	1.9 ± 0.06	0.005 ± 0.001	-5.84 ± 1.0	0.07%
CuO	0.41	4	1.94 ± 0.02	0.001 ± 0.002	-4.5 ± 1.2	0.11%
Rh-N-C		4.1 ± 0.6	1.99 ± 0.05	0.004 ± 0.002	-1.31 ± 1.8	0.26%
Rh ₂ O ₃	0.67	6	2.02 ± 0.02	0 ± 0.001	-0.57 ± 0.8	0.05%
La-N-C		8.2 ± 1.4	2.61 ± 0.3	0.03 ± 0.006	5.77 ± 1.0	1.74%
La ₂ O ₃	1.4	7	2.6 ± 0.2	0.015 ± 0.003	5.57 ± 0.6	0.56%
Ce-N-C		$0.35 \pm 0.1 /$ 5.5 ± 0.6	$1.65 \pm 0.7 /$ 2.60 ± 0.2	$0.00 \pm 0.04 /$ 0.015 ± 0.02	1.29 ± 5.7	0.91%
CeO ₂	0.70	8	2.27 ± 0.1	0.011 ± 0.004	5.97 ± 1.7	1.58%

Table S8. Fitting parameters for the EXAFS spectra $\chi(k)k^2$ in R-space and Fourier transform k range.

	R min (Å)	R max (Å)	kmin (Å ⁻¹)	k max (Å ⁻¹)
Cr-N-C	1	2	2	8
Mn-N-C	1	2	3	8
Co-N-C	1	2	3	8
Ni-N-C	1	2	3	8
Cu-N-C	1	2	3	8
Rh-N-C	1	2	3	8
La-N-C	0.8	2.5	3	8
Ce-N-C	0.8	2.5	3	8

Fe-N-C	1	2	3	7.3
--------	---	---	---	-----

8. WT-EXAFS should be compared with the standards.

Thank you to the reviewer for this point, we have performed WT-EXAFS on the M-N-C and corresponding metal oxide reference. This analysis has been added in Figure S18-S19

Figure S 18. FT-EXAFS and wavelet transformation (WT) for f-metals (a-c) La-N-C and the corresponding La₂O₃ standard and (d-f) Ce-N-C and the corresponding CeO₂ standard. In the WT-EXAFS, in addition to the peak intensity observed at ca. 2.1 Å at low k-values, an additional peak is present at ca. 3.7 Å again at low k-values, indicating interactions with only low Z-number neighbors, in particular La-C. For Ce-N-C, the WT-EXAFS shows two features located at ca. 1.5 Å and 2 Å, both located at low k-values (ca. 4-5 Å⁻¹), matching the Ce-O intensity in the CeO₂ WT-EXAFS, indicating the split peaks for Ce-N-C constitute interactions between Ce and low Z number neighbors (C, N or O atoms). While an additional feature at 3.7 Å is also observed at higher k-number (7 Å⁻¹), suggesting that in addition to atomically dispersed Ce-N_x sites, the formation of

small oxide clusters can coexist.

Figure S 19. EXAFS and wavelet transformations for (a) Fe-N-C and the corresponding Fe₃O₄ standard, (b) Rh-N-C and the corresponding Rh₂O₃ standard (c) La-N-C with the corresponding La₂O₃ standard.

9. The evaluated M-N coordination number in Table S3 was far higher than M-N4.

This is an important note raised by the reviewer. In some cases a higher coordination number was observed, in others a lower coordination, which has been addressed in supplementary note 2.

Supplementary note 3: EXAFS analysis - coordination of M-N-C catalysts

The observed coordination number for the M-N-C catalysts varies between ca. 1.8 to 8.4. While Rh-N-C displays a coordination number of 4 (that anticipated for M-N4 motifs), for Fe- and Cu-N-C, it is likely that in addition to the M-N bonds, some additional

M-O bonds are present, leading to a slightly increased coordination number. For Ni-N-C, the low coordination number could be attributed to the large structural disorder due to the off-center displacement of the Ni for the center of the Ni-N₄ moiety (as indicated by the XANES features). While for Ce-N-C, the low coordination number is due to a highly distorted local environment. Additionally, for the complex f-metals, La- and Ce-N-C, in addition to the M-N feature at low bond distance, the wavelet transformations of the EXAFS data also show an additional wave peak at higher R- and k-values, indicating small oxide clusters not detected by other techniques may co-exist.

10. Peak shifting can be observed in the XRD patterns (Figure S4).

The peak shift observed for the carbon peaks in the XRD pattern are likely a result of changes in the carbon morphology arising during the pyrolysis steps in the synthesis or intrinsic interaction of the metal with the carbon during pyrolysis (induced graphitization). It is known that amorphous carbon will show a peak around 22°, while a graphitic peak will show at a higher reflection (ca. 25°) as shown in the reference provided here.¹ Since our pyrolysis conditions (temperature and atmosphere) are optimized to maintain atomically dispersed sites, this can impact the carbon morphology.

1. [López-Romero, S., and J. Chávez-Ramírez. "Synthesis of TiC thin films by CVD from toluene and titanium tetrachloride with nickel as catalyst." *Matéria (Rio de Janeiro)* 12 (2007): 487-493.]

11. The XPS of Fe-N-C is present in both Figure S10 and Figure S11, the quality of the data was very low as compared to other M-N-C with similar metal loading.

The reviewer is correct with this observation of the low Fe 2p signal in the XPS spectra as compared to its other metal counterparts. This very low count Fe 2p signal is very common in iron based single atom catalysts and is often not fitted, but rather qualitatively shown in other literature. Our hypothesis for this low count signal is due to the high bond strength of the Fe-C bond which induced the Fe-catalyzed graphitization of carbon during pyrolysis which can lead to a significant amount of sub-surface Fe-N_xC_y species (in some cases, encased in a graphitic shell). This sub-surface Fe leads to a reduced XPS signal count. An example of this is clearly seen in our recent work in *Materials Today* in which almost no Fe 2p signal is initially observed in the catalysts, however, after Ar etching in the XPS to remove the surface layers, the Fe 2p signal is clearly observed, unveiling the sub-surface Fe species. Table S2 below shows the quantitative Fe 2p species as undetected before Ar etching and Figure 6b shows the same data in a graphic format.

Huang, Ying, et al. "Catalysts by pyrolysis: Direct observation of chemical and morphological transformations leading to transition metal-nitrogen-carbon materials." *Materials Today* 47 (2021): 53-68.

Table S2. XPS survey results for the precursor and samples pyrolyzed at different temperatures out in Ar environment with temperature ramp-rate of 10 °C min⁻¹.

	O 1s	C 1s	N 1s	Si 2p	Fe 2p 3/2
Precursor	42.7% (±1.05%)	27.3% (±1.38%)	3.8% (±0.03%)	26.0% (±0.37%)	0.24% (±0.02%)
435	18.8% (±0.20%)	60.8% (±2.48%)	10.4% (±1.37%)	9.8% (±0.91%)	0.17% (±0.03%)
870	21.0% (±0.40%)	62.1% (±1.39%)	5.1% (±0.55%)	11.7% (±0.59%)	0.22% (±0.02%)
1180	19.1% (±0.15%)	66.6% (±1.63%)	2.0% (±0.28%)	12.3% (±1.34%)	0
1180 Ar-etching	23.4% (±0.48%)	51.7% (±1.06%)	1.5% (±0.15%)	23.2% (±0.84%)	0.23% (±0.03%)

12. Line 152, “Figure 2d-e.” should be “Figure 2d-f”?

Thank you to the reviewer for catching this mistake, this change has been made.

13. Line 232 “NO3-” should be “NO2-”?

Thank you to the reviewer for catching this mistake, this change has been made.

14. The title of Table S5 should be the results of M-N-C rather than metal free samples?

Thank you to the reviewer for catching this mistake, this change has been made.

Reviewer #2

Comments:

The author prepared 13 kinds of single atomic catalysts with different elements, and applied them to the nitrate electroreduction. Through experimental analysis and theoretical calculation, the author proposed descriptors for evaluating the activity of M-N-C catalysts for the NO₃RR and NO₂RR. Unfortunately, there are many problems, and there is a lack of detailed explanation for supporting key conclusions. Therefore, I do not recommend publishing this paper at its present version. A revised might be reconsidered if the authors can address all the following concerns.

1. For single atomic catalysts, XAS test and fitting analysis are necessary, otherwise the results are not convincing. Although the HRTEM images can intuitively display the morphology of single atoms, there is a great contingency. XRD, XPS and EDS can not confirm the single atomic structure.

Updated XAS data with the corresponding k-space analysis and EXAFS fittings has been added (Figure S10-12 and Table S2 and S8 and full analysis of the XANES and EXAFS results in Supplementary note 1 and 3), please see the reply to Reviewer 1 (questions 6-8). We think there might be some confusion here as no HRTEM was used or mentioned in this manuscript. HRTEM cannot be directly or intuitively interpreted due to contrast reversal and focusing challenges. Aberration corrected scanning transmission electron microscopy AC-STEM was used to obtain atomic resolution images, visualizing the atomically dispersed sites. We agree with the reviewer that XRD and XPS alone can suggest but **does not** confirm the formation of single atom sites. EDS only shows the elemental distribution and was not employed in this study to claim any atomic dispersion of the metal sites. We employed low accelerating voltage aberration corrected and monochromated EELS to both visualize the atomically dispersed metal and detect simultaneously the metal and nitrogen edges of the atomically dispersed metal sites, suggesting a M-N_x coordination. We further utilized synchrotron based XAS, specifically EXAFS to show the local coordination environment, supporting the formation of atomically dispersed M-N_x sites.

2. The author thinks that the single atom structure should be M-N₄ (Fig. 1b). why do elements exhibit multiple valence states?

Thank you to the reviewer for raising this point, it is important to clarify this. We have been careful to explicitly state we have the confirmation of M-N_x sites in the catalyst. However, we are aware that a plurality of M-N_xC_y sites are present in all M-N-C catalysts (other than well-defined metal-phthalocyanines and certain metal organic frameworks), with the ideal M-N₄ site being among those.

3. For fig. 3b, why should HER be taken as the baseline and deducted? The analysis of “LSV with HER subtracted” for NO₃RR and NO₂RR is unreasonable. This treatment means that the authors believe that HER and NO₃RR processes coexist at a specific potential. However, the subsequent test results show that for some elements, HER almost does not occur or can be ignored. Obviously, this is very contradictory to the author's operation. Actually, in the solution where H⁺/H₃O⁺ and NO₃⁻ coexist, NO₃RR may be superior to adsorption or subsequent hydrogenation process. Thus, directly deducting HER is unreasonable.

The response here is the same as the response for Review 3 question 3.

In the initial submission, we were trying to use the HER subtracted LSV as a qualitative indicator of the activation of each reaction. But here, we totally agree with the reviewer regarding this point. Therefore, all HER subtracted LSVs were removed from the main text and ESI as follows.

- Figure 3b (b) LSV with HER subtracted from the NO₂RR/NO₃RR in Figure 3a
- Figure S22. Onset potential of the M-N-C (a) metal-free N-C (b) catalysts for the HER-subtracted from the NO₂RR/NO₃RR.
- Figure S24. HER subtracted linear sweep voltammetry in 0.01M KNO₂ (NO₂RR) and 0.16M KNO₃ (NO₃RR) for all metal free N-C and M-N-C catalysts.

These dropped figures were replaced with LSV's without HER subtraction, as detailed below. The drop of these figures does not change the conclusion of the manuscript.

In addition, as requested by reviewer #3, Figure 3c-d in original submission were replaced with standard NO₂RR and NO₃RR LSVs without HER subtraction, as shown by Figure 3 b and c in this revision. Additionally, Figure S27 looks at the onset potential for the NO₂RR/NO₃RR and Figure S28 has the raw LSV data obtained by RDE without HER subtraction.

Figure S 27. Onset potential of the (a) M-N-C and (b) metal-free N-C catalysts for the NO₂RR and NO₃RR. The onset potentials were determined at a current density of -1.0 mA/cm². Cr- and Cu-N-C exhibit the lowest onset potentials for the NO₃RR, while Fe-N-C demonstrates the lowest onset potential for the NO₂RR. Note that Ru-N-C exhibits an early onset potential for both the NO₃ and NO₂RR, however, the NO₃ and NO₂RR electrolysis results in Figure 3d and 4d suggest activity is largely attributed to the HER. Figure b suggest that over the metal free N-C catalysts the onset potential for the NO₃RR not impacted by the varying pyrolysis conditions, while for the NO₂RR, a slightly earlier onset potential is shown for the NO₂RR.

Figure S 28. Linear sweep voltammetry in 0.05M PBS (HER), 0.01M KNO_2 (NO2RR) and 0.16M KNO_3 (NO3RR) for all metal free N-C and M-N-C catalysts.

4. All the data provided in Supplementary Information should be reflected in the manuscript.

Thank you to the review for raising this point. We have gone through the manuscript and ensured that all Supplementary Figures, Tables and Notes have been represented/reported in the main manuscript.

5. If the FE of HER need to be discussed, the author should use gas chromatography for quantitative analysis. Whether the photos of bubble overflow is not valid evidences.

Thank you to the reviewer for bringing up this important point which needs more clarification. We agree that if any statement about the quantitative nature of the gas phase products will be employed, gas chromatography or mass spectrometry is necessary. However, we are not quantitatively considering the FE of gas phase products. Using the gap analysis plot in Figure 3e, we are simply showing the FE which goes to liquid phase products (NH_3 and NO_2^-) and the gap lets the reader quickly notice any efficiency that is lost to other gas phase products. The use of the photographs in Figure S37 (now Figure S41) are simply to note that bubbles were observed on the working electrodes during the NO_3RR electrolysis suggesting gas phase products (what the products are and exactly their corresponding FE was not quantitatively considered here).

6. The author should discuss in detail why $\text{Yield}_{\text{NO}_2\text{RR}-\text{NH}_3}$ and $\text{FEN}_{\text{O}_3\text{RR}-\text{NH}_3}$ are linear? What does a linear relation mean? The author lacks the necessary derivation process, What is the core of this work?

We thank the reviewer for raising this discussion and believe the reviewer was referring to Figure 4e-f. The core of this work is providing experimental support for a $2e^- + 6e^-$ reaction mechanism, wherein the NO_3RR is a cascade reaction with NO_2^- being an active reaction intermediates as follows.

Normally, in the field of the NO_3RR , the NO_3RR is assumed to follow a direct $8e^-$ pathway and any detected bulk NO_2^- is observed as an undesired byproduct. A direct $8e^-$ transfer pathway is particularly assumed when the ammonia selectivity is much higher than the nitrite selectivity (as observed by quantification in the bulk electrolyte). This assumption intuitively make sense when the minimum bulk NO_2^- is detected for these ammonia selective catalysts, and moreover because it is extremely challenging to experimentally demonstrate a $2e^- + 6e^-$ cascade pathway in this scenario and is therefore often assumed to be a direct $8e^-$ pathway. In our work, we demonstrate strong correlation between the NO_3RR ammonia selectivity and the NO_2RR activity, revealing a $2e^- + 6e^-$ pathway and active role of NO_2^- .

That is, for the NO_3RR cascade reaction, the catalysts with higher NO_2RR activity can effectively push the reaction to ammonia and minimize the bulk NO_2^- . Therefore, in the case of a 100% FE for nitrogen transformation with NH_3 and NO_2^- being the dominant products (most cases at low overpotentials in this work), there should be a strong correlation between the ammonia selectivity in the NO_3RR and the NO_2RR activity in parallel electrolysis. Such strong correlation over a wide range of M-N-C and N-C catalysts can even act on the highly ammonia-selective catalysts such as Fe-N-C wherein the NO_3RR FE_{NH_3} was near 100% and minimum bulk NO_2^- was detected, for which the $2e^- + 6e^-$ reaction mechanism cannot be experimentally revealed otherwise.

These correlations were initially developed to give a qualitative indication of the $2e^- + 6e^-$ cascade pathway for the NO_3RR , instead of telling any underlying numerical equation/derivation, since the electrochemical NO_3RR is a highly complex reaction with contributions from minor active sites (e.g., N-moieties in the M-N-Cs) and potential competition between NO_3^- and NO_2^- on the same active site. So, rather than giving a derivation with too many assumptions and simplifications that cannot be effectively validated, we have adjusted the wording by replacing the phrasing 'linear correlation' with 'strong positive correlation'. We kept the fitted line just to guide the eye.

“To examine the relationship between the NO_2RR and NO_3RR , Figure 4e and f correlates the activity of NO_2RR ($\text{FENH}_3 / \text{Yield}_{\text{NH}_3}$) with the selectivity of NO_3RR (FE_{NH_3}), revealing a strong positive correlation at -0.20 V and -0.40 V. “

7. Necessary explanations should be provided, such as “[NO_2RR YR NH_3] | [NO_2RR FE NH_3] vs. NO_3RR FENH_3 ”. Meanwhile, the author should explain why to do such complex operations, or just find a relationship?

We believe the reviewer was referring to Fig S49 in the original submission (and Fig S54 in current revision). The NO_2RR and NO_3RR referred to the reaction. YR and FE referred to yield rate and Faradic Efficiency, NH_3 or NO_2^- referred to the products. We made full explanations in the figure captions for those figures in ESI, but did not update the x and y labels in the figures. We thank the reviewer for pointing this out and we have updated those abbreviations in Figure S49-S52 (and Fig S54-S57 in current revision), making sure they are in consistent with their captions and figure 4e-g. For example, the “ NO_3RR FE NH_3 ” has been changed to “ NO_3RR , FE_{NH_3} ”, and “[NO_2RR YR NH_3] | [NO_2RR FE NH_3]” has been changed to “ NO_2RR , $\text{Yield}_{\text{NH}_3} / \text{FE}_{\text{NH}_3}$ ”.

We have been actively exploring multiple experimental descriptors to decipher the complex NO_3RR process, such as the NO_3RR selectivity towards NH_3 or NO_2^- , the activity of NO_2RR (yield, FE, total current). The complex “ NO_2RR , $\text{Yield}_{\text{NH}_3} / \text{FE}_{\text{NH}_3}$ ” is equal to the NO_2RR total current (NO_2RR , j_{total}). We expressed it in the dividend of Yield and FE, because we used yield rate and FE as the descriptor in other cases throughout the manuscript. Figure S54 and S55 showed either $\text{Yield}_{\text{NH}_3}$ or $j_{\text{NO}_2\text{RR}}$ had good and similar correlations with the NO_3RR FE_{NH_3} . The following note has been added to the caption of Figure 4e-f.

“The NO_2RR , $\text{Yield}_{\text{NH}_3} / \text{FE}_{\text{NH}_3}$ is equal to the NO_2RR total current (NO_2RR , j_{total}). The correlation between NO_2RR , $\text{Yield}_{\text{NH}_3}$ along and NO_3RR , FE_{NH_3} is shown in Figure S57”

8. For all single atomic catalysts provided by the author, the current density is very small

(the maximum current density is less than 20 mA cm⁻²). At present, the current density and FE of NO₃RR to ammonia are very high (FE>90%; current density up to 2 A cm⁻²). What is the significance or advantage of monatomic catalyst?

This is an important point raised by the reviewer. As noted in the manuscript, this is a more fundamental study to look at the intrinsic activity of the different M-N_x sites in a neutral electrolyte. However, we did show in Figure S26, that we could achieve NH₃ partial current densities up to 175 mA/cm² in an alkaline solution (at Fe = 0.6 wt%). However, as we will show in a work coming very soon an appealing application of the M-N-C material in the NO₃RR is not only as the primary catalyst, but as an active support replacing the inert Vulcan/Carbon Black often seen. As an example of this, we show here in Figure a & b below, a preview of this upcoming work. Figure. a below shows the layout of the material, where we have the metal nanoparticle supported on the M-N-C catalyst, while Figure. b below shows a durability study at ~1.3 A/cm² over 24 hrs (maintaining a FE_{NH₃} between 90-100% at a high yield rate). We find in this study that the M-N-C can enhance the NO₃RR activity through chemical interactions & providing more active sites, as well as increase the stability of the catalyst material.

9. According to Marcus-type theory (MPIP. J. Club-Mainz., 2008, 1, 1–13; Chem. Sci., 2013, 4, 2710–2723), the transfer of electrons occurs one by one. Why do you consider the first two steps together in the DFT calculations?

Reviewer is correct and when we study the mechanism of NO₃RR or NO₂RR (please refer to the answer to question 10 and our previous work on NO₃RR in ACS Catalysis 2022, 12, 11, 6651–6662), we do consider electron transfer to occur one by one. We “combined” reactions in the design of descriptors when we tried to find descriptors with

best correlation between the DFT calculated energetics and the experimentally determined activity and selectivity. Considering complex reaction mechanism of NO₃RR, energetics of “combined” reactions seem to have better correlation with the experiment.

10. For the mechanism of [$*+NO_3 \rightarrow *NO_3$], one electron is involved. For the mechanism of [$*+NO_3 \rightarrow *NO_2$], one electron and two protons are involved. When the above mechanisms are beyond the CHE model, how did you calculate these decoupled electron and proton energies?

For the descriptor [$*+NO_3 \rightarrow *NO_2$], we combined reactions [$*+NO_3 \rightarrow *NO_3$] involving one electron and the next step [$*NO_3 + 2H^+ + 2e^- \rightarrow *NO_2 + H_2O$] involving 2 proton/electron pair. The details of the computational procedure are explained in more detail in SI of our previous paper ACS Catalysis 2022, 12, 11, 6651–6662 <https://doi.org/10.1021/acscatal.2c01367>. Computational procedure is taken from previous work (F. Calle-Vallejo, *Phys. Chem. Chem. Phys.*, 2013, **15**, 3196-3202, <https://doi.org/10.1039/C2CP44620K>) and following works such as Niu et al., (*Adv. Funct. Mater.* 31, 2008533 (2022), <https://doi.org/10.1002/adfm.202008533>). In short, decoupled electron/proton reaction or [$*+NO_3 \rightarrow *NO_3$] reaction was modeled using Hess law to avoid calculations involving solvated NO₃⁻ (to avoid errors in the DFT description of solvated NO₃⁻).

Figure is taken from F. Calle-Vallejo, *Phys. Chem. Chem. Phys.*, 2013, **15**, 3196-3202.

Adsorption energies [$*+NO_3 \rightarrow *NO_3$] (labeled ΔG_{NO_3} in the Figure) were related to those calculated with DFT using H⁺/e⁻ “coupling” (labeled ΔG_{ADC}^{DFT} in the Figure) through $\Delta G_{NO_3} = \Delta G_{ADS}^{DFT} + 0.075 + 0.317$. Values of 0.075 and 0.317 eV are experimental values. Similar schematic was used for [$*+NO_2 \rightarrow *NO_2$] with corresponding experimental values for HNO₂(aq) and HNO₂(g).

11. For the oxidative associative adsorption mechanism, [$*+NO_3^- \rightarrow *NO_3$], if the electron energy is $-eU$, the Gibbs free energy of this step increases along with decreasing potential. It is well known that the negative potential corresponds to better performance in the reduction reaction. How do you consider this question?

Reviewer is certainly right, and this is something to consider. We discuss this effect in our previous paper when discussing the cell potential and conditions that prefer $*NO_3$ or $*NO_2$

adsorption over $*H$ adsorption (Figure S10 in SI of our previous paper ACS Catalysis 2022, 12, 11, 6651–6662). We also found only one other paper discussing change in Gibbs free energy with applied cell potential of different NO_3RR steps that involve oxidation, reduction, or no electrons. This is a very interesting point that we decided not to discuss to maintain the focus of the paper on the library and performance of different M-N-C catalysts in NO_3RR/NO_2RR (in addition, we discuss this in our previous paper).

Figure taken from Lim et al, ACS Catalysis 11, 7568 (2021)

<https://doi.org/10.1021/acscatal.1c01413> and shows how energy for NO_3^- associative and dissociative adsorption and NO_2^- desorption and NO_2^- conversion to $*NO + *O$ on Pd catalyst change with applied potential

12. Line 308, you have lost a minus symbol ($\Delta rG \sim -4$ eV). It is very troublesome in rigorous scientific research.

Thank you to the reviewer for catching this important point. We have made the correction.

13. In the manuscript figure 5(a), the author said that the Mo-N4, La-N4, Ce-N4 and W-N4 sites could strongly adsorb NO_3^- in a dissociative manner, forming an $*O$ surface species and a free NO_2^- molecule. However, the La-N4 and Ce-N4 perform the competition of $*NO_2$ and $*NO_3$. Meanwhile, the author said that the Cr-N4, Mn-N4, Fe-N4, and Ru-N4 sites, associative and dissociative adsorption of NO_3^- is likely a competitive process that also depends on the cell potential. However, the Cr-N4 prefers

to form an *O surface species and a free NO₂- molecule. In addition, the conclusion that Rh-N₄ has poor NO₃RR activity from the DFT descriptors is not rigorous.

La-N₄ and Ce-N₄ perform the competition of *NO₂ and *NO₃ if NO₂⁻ formed can re-adsorb on the surface. We emphasize that “new active site (O-M-N₄) can still coordinate a NO₃⁻/NO₂⁻ molecule for further reduction”.

For Cr-N₄, Mn-N₄, Fe-N₄, and Ru-N₄ both associative and dissociative adsorptions are competitive. As one is potential dependent (associative adsorption) and the other is not potential dependent (dissociative adsorption), the competition between the two processes will be potential dependent. Based on DFT calculations of elementary step energetics, Cr-N₄ would mainly adsorb NO₃⁻ dissociatively.

To explain this in more detail we added

For the Cr-N₄, Mn-N₄, Fe-N₄ and Ru-N₄ sites, associative and dissociative adsorption of NO₃⁻ is likely a competitive process that also depends on the cell potential. Namely, while associative adsorption of NO₃⁻ is potential dependent, dissociative adsorption is not as it does not involve electrons.

We have softened the tone regarding firm conclusions to the Rh-N₄ sites and put this in perspective with other NO₃RR computational literature and our experimental observations. Additionally, our DFT descriptors suggest limited NO₃RR activity over Rh-N₄ sites, being only favorable via the reductive adsorption of NO₃⁻, while in other computational based works, Rh-N₄ sites suffer some competition between *H and *NO₃, in agreement with the poor NO₃RR activity observed in our experimental observations. More understanding can be deduced from Figure 5b.

Wang, Yian, and Minhua Shao. "Theoretical Screening of Transition Metal–N₄-Doped Graphene for Electroreduction of Nitrate." *ACS Catalysis* 12.9 (2022)

Wang, Shuo, et al. "High-throughput identification of highly active and selective single-atom catalysts for electrochemical ammonia synthesis through nitrate reduction." *Nano Energy* 100 (2022)

At this point we also want to emphasize that NO₃RR (and NO₂RR) are very complex processes and we tried to emphasize the importance of this fact in the design of active and selective catalysts. This is especially true when considering associative and dissociative adsorption of NO₃⁻ and possible desorption of *NO₂ intermediate and its readsorption (at different conditions for example). Calculations can complement experiments and provide key details regarding these processes and their probability on different M-N-C catalysts. We find that Figure 5b is especially valuable in this respect.

14. The author calculated the Gibbs free energy at experimental pH of 6.3, but the experimental condition is pH=14. The influence of pH may need to be considered to deepen the mechanism, as the nitrite/nitrate reduction showed pH dependence (Chem. Commun., 2014, 50, 2148-2151; J. Am. Chem. Soc., 2018, 140, 2012-2015).

We apologize for the confusion. Local pH is 6.3; therefore, correlation was made for the right experimental pH. We used pH=14 only one experiment using Fe-N-C in the ESI to show high current density achieved for the alkaline condition. To address this, we emphasized the neutral pH of 6.3 several times throughout the text (highlighted in the changes to the manuscript file).

15. The solution effect is very important in electrochemistry, but you didn't use any solution correction in the DFT calculations. In addition, the spin effect was not taken into account.

We agree with the reviewer that solvation might be important; however, we do not expect this to change relative adsorption energies of NO₃RR/NO₂RR intermediates on different sites. Previous works of Skúlason, Nørskov, and others showed that the key interactions between adsorbed species and solvent are very similar for the same adsorbed species and effect of solvation can be estimated from the information for hydrogen bonds in different *O–H or *N–H species adsorbed on any surface [E Skúlason, et al Phys. Chem. Chem. Phys. 14 (2012) 1235–1245, doi:10.1039/C1CP22271F.]. Therefore, we do not expect solvation to change the observed DFT trends and correlations.

Spin effect was considered by performing spin polarized calculation (ISPIN=2 option in VASP). This is an extremely important aspect of the calculations as different metals are in different spin states, which changes the way they interact i.e., adsorb different reactants/intermediates. To address this, we added a sentence in computational details:

All calculations were performed as spin polarized calculations.

Reviewer #3

Comments:

The work by Murphy et al. is an extensive consideration of M-N-C electrocatalysts for NO₃RR, targeting NH₄⁺ in neutral media. Considering a wide range of TM centers, the authors thoughtfully probe the role of the NO₂⁻ intermediate and a potential cascade reaction through an intriguing isotopic labelling study, with complementary consideration of different modes of adsorption of NO₃⁻ and NO₂⁻ by DFT. The topic is timely and of interest to the readers of Nature Communications. Although the work is thorough and

compelling, there are a couple points which must be addressed before publication:

1. Introduction: Page 2 line 62, "Previous attempts..." It seems remiss to not put this work in context with recent work considering competitive adsorption of H and NO₃ on the surface of 3d and 4d transition metals in neutral media and implications regarding NO₃RR FE and selectivity to NH₄⁺ (<https://doi.org/10.1021/jacs.2c05673>).

Thank you to the reviewer for bringing this very nice recent work to our attention. The paper study suggested here provides a very relevant discussion to our work in terms of extended metal surfaces in contrast with atomically dispersed sites in this work, there are interesting contrasts and similarities, ultimately providing NO₃RR catalyst development parameters. We have incorporated a discussion of this work into our introduction section as follows.

"A recent complementary work over a series of extended transition metal surfaces (in contrast to the atomically dispersed M-N_x sites in the present study) by Carvalho et al. developed NO₃RR catalyst design parameters by linking the electronic structure to experimentally observed NO₃RR performance. This work derived microkinetic models that successfully described rate limiting step (NO₃⁻ to NO₂⁻) and further the NO₃RR FE_{NH₃} by developing microkinetic models, identifying a competitive adsorption between protons and nitrate, well described by the material-dependent $\Delta G_{H^*} - \Delta G_{NO_3^-}$. The study found that increasing FE_{NH₃} correlates with NO* binding and subsequent dissociation into N* and O* as the d-band energy approaches the Fermi energy, resulting in the Co-foil showing the highest ammonium selectivity."

2. Figures 2 j-l, S10-14, and discussion of XPS spectra: In general, the author's description of XPS data collection (incident X-ray source and energy, use of charging neutralization), binding energy scale calibration, sufficiently wide binding energy window collected to observe background regions and corresponding background selections, and fitting procedures and justifications are lacking in both literature support and discussion. This could be provided in the SI.

Thank you to the reviewer for suggesting the need for clarification in the XPS acquisition and fitting. The following discussion has been added to the experimental section SI.

The XPS was performed on a Kratos AXIS Supra spectrometer with a monochromatic Al K α source (with an emission current of 15 mA and X-ray power of 225 W, while for the high-resolution spectra the emission current and power are 20 mA and 300 W, respectively). No charge neutralization was employed as all these samples are highly conductive, carbon-based catalysts. Survey spectra were obtained at a pass energy of 160 eV from 1400 eV to 5 eV at a step size of 1 eV. The High-resolution C 1s, N 1s and O 1s spectra were obtained at a pass energy of 20 eV with a 0.1 eV step size. The metal spectra were obtained at a pass energy of 40 eV with a 0.1 eV step size. A C 1s spectra was also obtained at a pass energy of 40 eV for calibration. CasaXPS software was used to analyze and fit all spectra and all spectra were calibrated based on the sp²

carbon (284 eV). A Linear background was employed for the C 1s and N 1s spectra, while a Shirley background was used for the O 1s and metal spectra. The sp^2 carbon in the C 1s spectra was fit with a 80% Gaussian / 20% Lorentzian function and all other spectra were fit using a 70% Gaussian / 30% Lorentzian function. All fitting parameters employed in this work are based on previous fittings from our group that are reported in the literature references here for this class of M-N-C catalysts.^{3,4,5,6}

[3] Artyushkova, Kateryna, et al. "Chemistry of multitudinous active sites for oxygen reduction reaction in transition metal–nitrogen–carbon electrocatalysts." *The Journal of Physical Chemistry C* 119.46 (2015): 25917-25928.

[4] Artyushkova, Kateryna, et al. "Oxygen binding to active sites of Fe–N–C ORR electrocatalysts observed by ambient-pressure XPS." *The Journal of Physical Chemistry C* 121.5 (2017): 2836-2843.

[5] Matanovic, Ivana, Kateryna Artyushkova, and Plamen Atanassov. "Understanding PGM-free catalysts by linking density functional theory calculations and structural analysis: Perspectives and challenges." *Current Opinion in Electrochemistry* 9 (2018): 137-144.

[6] Artyushkova, Kateryna. "Misconceptions in interpretation of nitrogen chemistry from x-ray photoelectron spectra." *Journal of Vacuum Science & Technology A: Vacuum, Surfaces, and Films* 38.3 (2020): 031002.

3. Pg. 9 line 202 & Figure 3b "HER subtracted LSV..." Subtracting HER current as a "background" current implies that the rate of HER is unchanged by the addition of NO_3^- (presumably leading to competitive adsorption between H and NO_3^- on the surface, lacking discussion in the present work), which does not seem supported by the high NO_3RR FE generally observed. Please present the LSV without this subtraction, overlaying HER if a comparison is desired.

In the initial submission, we were trying to use the HER subtracted LSV as a qualitative indicator of the activation of each reaction. But here, we totally agree with the reviewer regarding this point. Therefore, all HER subtracted LSVs were removed from the main text and ESI as follows.

- Figure 3b (b) LSV with HER subtracted from the NO_2RR/NO_3RR in Figure 3a
- Figure S22. Onset potential of the M-N-C (a) metal-free N-C (b) catalysts for the HER-subtracted from the NO_2RR/NO_3RR .
- Figure S24. HER subtracted linear sweep voltammetry in 0.01M KNO_2 (NO_2RR) and 0.16M KNO_3 (NO_3RR) for all metal free N-C and M-N-C catalysts.

These dropped figures were replaced with LSV's without HER subtraction, as detailed below. The drop of these figures does not change the conclusion of the manuscript.

In addition, as requested by reviewer #3, Figure 3c-d in original submission were replaced with standard NO₂RR and NO₃RR LSVs without HER subtraction, as shown by Figure 3 b and c in this revision. Additionally, Figure S27 looks at the onset potential for the NO₂RR/NO₃RR and Figure S28 has the raw LSV data obtained by RDE without HER subtraction (these Figures and captions have been provided in the response to Review 2 question 3).

4. Fig 3 and methods: FE and selectivity data are based upon CA holds for a fixed amount of time (2h), both across samples and applied potentials. However, the authors (and others in literature) clearly articulate that NO₂⁻ is in many cases an intermediate, whose abundance in a batch cell will inherently depend upon the extent of conversion, which is uncontrolled via electrolysis time. The authors should present the extent of conversion of NO₃⁻ for different catalysts and the applied potentials considered, and if these differ greatly comment on the implications in conclusions made.

Thank you to the reviewer for this comment, this is an interesting point. However for the purpose of this more mechanistic study in these neutral conditions (not meant for highest current density performance, see Reviewer 2 Question 8) we have chosen a pseudo infinite concentration of NO₃⁻ (0.16M) such that with the mildest cathodic potential or strongest cathodic potential, the percent conversion (as defined as $([\text{NO}_3^-] (\text{initial}) - [\text{NO}_3^-] (\text{final})) / [\text{NO}_3^-] (\text{initial}) * (100\%)$) is always less than 1% for each catalyst under all conditions.

5. Fig 4b line 285 and pg 15 line 359 state that Ru and Rh are excluded from fitted correlations because of gaseous products (H₂ based on verbiage in S37), and Co as well (perhaps because of gaseous products pre line 359, but not detailed in the figure caption). The implications of this exclusion should be discussed in more detail. It seems disingenuous to call these “outliers” excluded from fits correlating interesting ties between NO₂⁻ and NO₃⁻, but then in the conclusion (lines 389-391) claim “a universal contribution from the NO₂⁻ intermediate” and that DFT calculations of NO₃⁻ and NO₂⁻ adsorption “explain the selectivity for each metal center.” The authors should be more clear about how far their conclusions go, their limitations (perhaps validity only in cases with near 100% FE to NO₃RR?), and the assumptions that go into them.

Thank you to the reviewer for bringing more light to this discussion, this is a very important point. Exactly as the reviewer points out, based on the very poor experimental NO₃RR performance of Ru and Rh (Figure 3e) and observation of gaseous products forming on the electrodes, these two metal centers are considered as outliers as these are the only two metal centers showing this unique behavior and showing the formation of gaseous products. The exclusion of the Co metal center is rationalized through its unique NO₃/NO₂RR performance as evidenced by its NO₃RR performance in Figure 3 and its performance in the NO₃RR with ¹⁵NO₂⁻ doping (Figure S44). As observed in Figure 3, Co-N-C has difficulty in activating NO₃⁻, resulting in poor FE. However, in the ¹⁵NO₂⁻ doping experiments (16,000 ppm ¹⁴NO₃⁻ with 10 ppm ¹⁵NO₂⁻), Co-N-C shows

significant activation of the NO_2^- while remaining relatively inert to the NO_3^- as seen in the ^1H NMR product spectra with the isotopic $^{15}\text{NH}_3$ doublet dominating the $^{14}\text{NH}_3$ triplet, indicating its unique and outlying behavior for the NO_3RR . The claim of the universal NO_2^- contribution is applicable to all M-N-C materials, regardless of the metal center. It was shown in the isotopic doping experiments that even over metal free N-C, these different nitrogen moieties present in all M-N-C materials can adsorb and reduce the $^{15}\text{NO}_2^-$ to $^{15}\text{NH}_3$ (even when $^{15}\text{NO}_2^-$ is present at very small concentrations compared to the $^{14}\text{NO}_3^-$). The correlation that is being referred to in lines 389-391 is not the computational correlation but the experimental-experimental correlation in Figure 4 e-f. We have changed the wording to make this clearer. The reviewer is correct in that we should be clearer with the limitations of these computation-experiment correlations. We have changed the wording in the main text in accordance with this.

"While these computational-experimental activity descriptors provide strong correlations for the NO_3RR and NO_2RR activity, these correlations are limited when the H^+ adsorption is a strongly competing factor leading to the HER over the NO_3RR (Ru- and Rh-N-C) and when there is poor NO_3^- activation in contrast with significant NO_2^- activation (Co-N-C)."

6. Pg 21 line 519: although the authors have previously shown that M-N-C catalysts are not active to reduce N_2 , does the presence of N_2 shift equilibrium potentials (and resultant current densities or product distributions) of NO_3^- reduction processes in comparison to e.g. saturating with Ar?

We thank the reviewer for raising this interesting point. As the reviewer correctly points out, we have established that these M-N-C catalysts are not active in reducing N_2 . To evaluate the impact of N_2 (vs. Ar) on the equilibrium potentials, we have performed open circuit voltage measurements with N_2 saturation and Ar saturation in the 0.05M PBS + 0.16M KNO_3 electrolyte, as shown below. From this figure, there is a negligible impact when purging either N_2 or Ar in the cell, therefore we believe that purging the cell with either N_2 or Ar does not alter the equilibrium potentials and resulting current densities. This figure has been added as Figure S63 to provide clarity for the readers and is referred to in the methods section.

Figure S 63. Open circuit voltage (OCV) measurements when the electrolyte is saturated in N₂ or Ar. The negligible difference in the OCV suggests that the use of either N₂ or Ar does not alter the equilibrium potentials of the NO₃RR process.

And a few general comments:

7. At the neutral conditions here the species formed from NO₃⁻ reduction should be NH₄⁺ (ammonium) as pH 7 is below the pKa of ammonia. While the detection method of the Berthelot reaction is robust for its detection, the nomenclature is important due to the implications on proton demand and the product remaining dissolved in solution.

We thank the reviewer for this comment on the distinction between the nomenclature of ammonia/ammonium. The reviewer is absolutely correct, at the pH of 6.5, we are below the pka of ammonia, and it should be denoted as the protonated ammonium NH₄⁺, which is highly soluble in aqueous electrolytes. As noted correctly by the reviewer NH₄⁺ is detectable by the Berthelot method which relies on the use of 1M NaOH, which is strong enough to transform the pH of our 0.05M PBS electrolyte to alkaline, which causing the NH₃/NH₄⁺ present to be in the NH₃ form. We agree this distinction in the nomenclature is very important and have clarified this in the manuscript that at our experimental conditions we are indeed reducing NO₃⁻ to NH₄⁺ but have left the discussion in terms of ammonia as the quantified product. The following has been discussion has been added into the manuscript for clarification.

“Note that at the pH = 7 conditions utilized in this work, the pka of ammonia has not been reached (pka 9.2), therefore the ammonia produced is in the protonated form ammonium (NH₄⁺). However, in the alkaline conditions present during the Berthelot method for detection, the NH₃/NH₄⁺ is present in the NH₃ form. Therefore, the discussion in this manuscript is for the detected product, ammonia (NH₃).”

8. Error bars are throughout the main text, but the source of them is not clearly identified in the manuscript or SI.

Thank you to the reviewer for raising this point, all the error bars in this work come from triplicate experiments and are then calculated using a 90% confidence level. A note has been added in the method section to make this point clearer for the reader.

9. What is the justification for the authors using geometric surface area for current density normalization and ammonia/ammonium yield calculations?

We thank the reviewer for this very relevant question. The use of geometric surface area for normalizing the current density and corresponding yield calculations was chosen to simply provide a way to compare the normalized yield rates among the various M-N_x active sites within this study. As the same electrode preparation and catalyst loading on the electrode was used for all electrochemical tests and the same synthesis approach is utilized for all catalysts the corresponding catalyst surface areas and porous structure are similar, such that we aim to compare the intrinsic activity across the selected metal centers on a geometric basis. We acknowledge that a more meaningful comparison and normalization for atomically dispersed catalysts is that of turnover frequency (TOF). However, this is extremely difficult and convoluted as the quantification of active sites in M-N-C catalysts is something that has only been pseudo-established for Fe-N-C catalysts (nitrite stripping, cryo-CO₂ adsorption, and recent FT-alternating current voltammetry) and is still something that is debated in field.

10. How has solution and contact resistance been accounted for?

Thank you for raising this important point, solution and contact resistances were not corrected for in this work and this will be noted in the experimental section to make this clear. One nice point of using 0.16M KNO₃ is that as it dissociates into K⁺ and NO₃⁻ ions, this creates a strong electrolyte with a high conductivity, largely reducing solution resistance. PEIS measurements in our system yield solution and contact resistances of ca. 13.5 Ω.

11. When performing two-cell selectivity measurements, are solution and contact resistances accounted for during the measurement? If not, what are the implications on the chronoamperometric potentials measured?

Both the linear sweep voltammetry and chronoamperometric measurements in this work are performed without iR correction. This is an important point and we have added a sentence in the experimental section to make this clear. As mentioned in the point above, an additional benefit of the large 0.16M KNO₃ concentration (in addition to the pseudo infinite NO₃⁻ concentration) is the significant electrical conductivity it provides to the electrolyte, minimizing the solution resistance. Albeit there remains a small iR

contribution with an impact of *ca.* -0.002V at -0.20V up to *ca.* -0.057V at -0.80V (all V vs. RHE), which at the lower applied potentials should have essentially no impact on the electrolysis performances. At the higher applied potential of -0.80V vs RHE (non-corrected), the lack of *iR* correction could slightly impact the chronoamperometric potential, however, at this high overpotential the reaction is already being forcefully driven such that the lack of *iR* correction does not alter any conclusion made in this study.

REVIEWER COMMENTS

Reviewer #1 (Remarks to the Author):

The manuscript was revised according to the comments provided. However, the impact of the $2e+6e$ mechanism was somewhat less pronounced as compared with the overall activity in the NO₃RR. In fact, the NO₃RR activity in alkaline conditions was by orders higher than that under neutral conditions, making the fundamental investigations under neutral conditions less important. In addition, the NO₃RR was only the side reaction of HER in this case. This issue has not been properly investigated in the theoretical calculation part. All in all, such an article is still far from the level of Nature Communications.

Other points:

1. Their theoretical models could not appropriately describe the different oxidation states of different metal centers. In principle, the number of unpaired electrons and the formal valence state of d state metal were non-related.
2. The perfect linear relationship can be observed between y_1 and y_2 axis in Figure 5b (Figure S60 b, c), did this implied something?
3. The trend of the catalytic performance shown in Figure 3d differed from the LSV result in Figure S28.
4. The structure of these M-N-C catalysts after the catalysis test should be checked.
5. The title/unit in many figures was wrong, such as in Figure 4e, Figure S29, S32, and S54-S57.

Reviewer #2 (Remarks to the Author):

The authors provide a suitable response and revision to the reviewers' concerns. Thus, this manuscript can be accepted at its present version.

Reviewer #3 (Remarks to the Author):

The authors have thoroughly addressed all my concerns and comments, as well as those from the other reviewers, and the work is greatly improved as a result.

This work will contribute valuably to the field of NO₃RR and be of great interest to the readership of Nature Communications.

Reviewer #1 (Remarks to the Author):

The manuscript was revised according to the comments provided. However, the impact of the $2e+6e$ mechanism was somewhat less pronounced as compared with the overall activity in the NO₃RR. In fact, the NO₃RR activity in alkaline conditions was by orders higher than that under neutral conditions, making the fundamental investigations under neutral conditions less important. In addition, the NO₃RR was only the side reaction of HER in this case. This issue has not been properly investigated in the theoretical calculation part. All in all, such an article is still far from the level of Nature Communications.

We thank the reviewer for recognizing our **proposed $2e^- + 6e^-$ pathway** for electrocatalytic NO₃RR over M-N-C catalysts. It is of particular value to us as at present the available publications in the field usually ambiguously assume a **direct $8e^-$ pathway** with NO₂⁻ as a leaked side product. This is partially due to the difficulty to experimentally investigate the underlying cascade reaction on ammonia selective catalysts. Indeed, it is of critical significance in academic research to decipher the intrinsic reaction mechanism of a complex reaction (such as NO₃RR) that is being widely studied now, but perhaps misunderstood by many, a fact that largely impacts both fundamental and applied science. Practically speaking, revealing the role of nitrite intermediate can help with eliminating the otherwise hard-to-separate NO₂⁻ when collecting ammonia as final product and results in developing more efficient catalytic systems, with tailored active sites to address specific reaction steps in the cascade.

The aim of this work is not to target maximum performance of the M-N-C catalysts for the NO₃RR. One key aim of this work is to evaluate and develop a set of **physically relevant activity descriptors**, evaluating the intrinsic activity of each metal center and revealing the reaction mechanism. As compared to alkaline conditions, a circum-neutral chemical environment can, to the larger extent reflect the intrinsic activity and selectivity of each active site, particularly at low overpotentials, making the electrochemical performance of each catalyst more distinguishable when the metal centers dominate the reactions.

Also, this work paved the road for leveraging the rich pool of M-N-C materials as either catalysts or active supports. For example, one of our ongoing works is utilizing the M-N-C as synergistic support to perfect the selectivity of alkaline NO₃RR under high current throughput, given the fact that alkaline NO₃RR is more active but still with imperfect ammonia selectivity.

Lastly, we do not quite understand the reviewer's argument on "NO₃RR was only the side reaction of HER". Because when looking at the large data set in Figure 3d, most cases showed a near **100% nitrogen transformation** with the FEs of NO₂⁻ and NH₃ adding up to ~100%, with only Ru, Rh and Co as the outliers that has been discussed in the manuscript. In a case where the main reaction accounts for nearly all the total observed current, this cannot be deemed a side reaction. Additionally, in no reported NO₃RR literature of similar or even lower NO₃RR selectivity is the NO₃RR termed as the side reaction of the HER.

1. Their theoretical models could not appropriately describe the different oxidation states of different metal centers. In principle, the number of unpaired electrons and the formal valence state of d state metal were non-related.

We thank the reviewer for pointing out this concern as it is extremely important to critically evaluate agreements between the models used for the computational study and the realistic materials tested in the experiments. Our team has extensive expertise in modeling M-N-C catalysts and one of the co-authors has recently written a computational tutorial addressing some of the best practices and challenges in modeling M-N-C catalysts for water splitting and fuel cell applications [S. Tosoni, G. Di Liberto*, I. Matanovic*, G. Pacchioni, Modelling single atom catalysts for water splitting and fuel cells: A tutorial review, J. Power Sources, 556, 232492 (2023) <https://doi.org/10.1016/j.jpowsour.2022.232492>].

We approached modeling of M-N₄ defects in M-N-C catalysts by creating two carbon atom vacancies in the graphene in which we placed a metal atom (not metal ions).¹⁻⁵ Structures were then optimized as neutral using unrestricted open shell spin density and with careful evaluation of the effect the initial magnetic moment has on the self-consistent solution of the electronic problem.^{6,7} Adsorption energies are given relative to the M-N₄ defect with magnetic solution with the lowest electronic energy.

However, we must **emphasize that M-N-C catalysts are challenging to simulate, partially due to their nature that lies at the boundary between homogeneous and heterogenous catalysts. Like homogeneous catalysts, metal atoms in M-N-C catalysts are individual atoms, whose chemical reactivity directly relates to their electron configuration**; however, just **like heterogeneous catalysts** metal atoms incorporated in graphene **have a strong interaction to the underlying support, which in turns influences their charge and spin state (i.e., oxidation state)**. **For that** reason, **we emphasize that** it is difficult to assign formal oxidation state of a single atom metal incorporated into the system with highly delocalized electrons, such as graphene. Calculated numbers will also depend on the adopted functional, which affects the degree of localization of charges and consequently the amount of charge on a metal atom in M-N-C catalysts.

To address reviewer's concerns, we included more details concerning our approach in modeling of metal-containing defects in M-N-C catalysts:

"We approached modeling of M-N₄ sites in the M-N-C catalysts by creating two carbon atom vacancies in the graphene in which we place a metal atom.¹⁻⁵ Structures were then optimized as neutral using unrestricted open shell spin density and with careful evaluation of the effect the initial magnetic moment has on the self-consistent solution of the electronic problem.^{6,7} Adsorption energies are given relative to the M-N₄ defect with total magnetic moment corresponding to the lowest electronic energy."

We also included following statement in the main manuscript explaining the limitations of the DFT work.

"We also emphasize that M-N-C catalysts are challenging to simulate, and the reliability of the computational results depends on many factors (e.g., see references^{8,9} and SI note 4). Although one should pay attention to the accuracy of the absolute numbers, relative values and trends are usually more reliable.

In addition, to emphasize the challenges and limitations of DFT modelling, we included following discussion in Supporting Information, note 4.

Supplementary note 4: Limitations of the computational approach and the difficulties in modelling M-N-C catalysts

M-N-C catalysts are challenging to simulate, and the reliability of the computational results depends on many key-factors. For example, parameters used in the Density Functional Theory (DFT) calculations such as number of k-points, spin polarization, dispersion correction, and choice of the exchange-correlation functional can affect computational results.⁹ In particular, there is evidence of deviations for high spin complexes when comparing results obtained using standard generalized gradient approximation and self-interaction corrected functionals. In addition, modeling of M-N-C catalysts represents a challenge as these materials lie at the boundary between homogeneous and heterogenous catalysts. Like homogeneous catalysts, metal atoms in M-N-C catalysts are individual atoms, whose chemical reactivity directly relates to their electron configuration. However, similarly to heterogeneous catalysts, metal atoms incorporated into graphene have a strong interaction to the underlying support, which in turns influences their oxidation state. Oxidation state will also depend on the adopted functional, which affects the degree of electron density localization and consequently the amount of charge on a metal atom in the M-N-C catalysts. Moreover, the real operating conditions in electrocatalysis are far

from the vacuum conditions assumed in our DFT calculations and a realistic modelling should account for solvent effects. Accounting for all these effects is extremely challenging⁸; however, interpretation of relative values and trends while cross-validating DFT results with experimental findings still provide reasonable and valuable insight.

- (1) Asset, T.; Garcia, S. T.; Herrera, S.; Andersen, N.; Chen, Y.; Peterson, E. J.; Matanovic, I.; Artyushkova, K.; Lee, J.; Minteer, S. D.; Dai, S.; Pan, X.; Chavan, K.; Calabrese Barton, S.; Atanassov, P. Investigating the Nature of the Active Sites for the CO₂ Reduction Reaction on Carbon-Based Electrocatalysts. *ACS Catal.* **2019**, 9 (9), 7668–7678. <https://doi.org/10.1021/acscatal.9b01513>.
- (2) Rojas-Carbonell, S.; Artyushkova, K.; Serov, A.; Santoro, C.; Matanovic, I.; Atanassov, P. Effect of PH on the Activity of Platinum Group Metal-Free Catalysts in Oxygen Reduction Reaction. *ACS Catal.* **2018**, 8 (4), 3041–3053. <https://doi.org/10.1021/acscatal.7b03991>.
- (3) Kodali, M.; Santoro, C.; Serov, A.; Kabir, S.; Artyushkova, K.; Matanovic, I.; Atanassov, P. Air Breathing Cathodes for Microbial Fuel Cell Using Mn-, Fe-, Co- and Ni-Containing Platinum Group Metal-Free Catalysts. *Electrochim. Acta* **2017**, 231, 115–124. <https://doi.org/10.1016/j.electacta.2017.02.033>.
- (4) Matanovic, I.; Artyushkova, K.; Atanassov, P. Understanding PGM-Free Catalysts by Linking Density Functional Theory Calculations and Structural Analysis: Perspectives and Challenges. *Curr. Opin. Electrochem.* **2018**, 9, 137–144. <https://doi.org/10.1016/j.coelec.2018.03.009>.
- (5) Sebastián, D.; Serov, A.; Matanovic, I.; Artyushkova, K.; Atanassov, P.; Aricò, A. S.; Baglio, V. Insights on the Extraordinary Tolerance to Alcohols of Fe-N-C Cathode Catalysts in Highly Performing Direct Alcohol Fuel Cells. *Nano Energy* **2017**, 34 (February), 195–204. <https://doi.org/10.1016/j.nanoen.2017.02.039>.
- (6) Li, J.; Sougrati, M. T.; Zitolo, A.; Ablett, J. M.; Oğuz, I. C.; Mineva, T.; Matanovic, I.; Atanassov, P.; Huang, Y.; Zenyuk, I.; Di Cicco, A.; Kumar, K.; Dubau, L.; Maillard, F.; Dražić, G.; Jaouen, F. Identification of Durable and Non-Durable Fe_{Nx} Sites in Fe–N–C Materials for Proton Exchange Membrane Fuel Cells. *Nat. Catal.* **2021**, 4 (1), 10–19. <https://doi.org/10.1038/s41929-020-00545-2>.
- (7) Mineva, T.; Matanovic, I.; Atanassov, P.; Sougrati, M. T.; Stievano, L.; Clémancey, M.; Kochem, A.; Latour, J. M.; Jaouen, F. Understanding Active Sites in Pyrolyzed Fe-N-C Catalysts for Fuel Cell Cathodes by Bridging Density Functional Theory Calculations and 57Fe Mössbauer Spectroscopy. *ACS Catal.* **2019**, 9 (10), 9359–9371. <https://doi.org/10.1021/acscatal.9b02586>.
- (8) Tosoni, S.; Di Liberto, G.; Matanovic, I.; Pacchioni, G. Modelling Single Atom Catalysts for Water Splitting and Fuel Cells: A Tutorial Review. *J. Power Sources* **2023**, 556 (November 2022), 232492. <https://doi.org/10.1016/j.jpowsour.2022.232492>.
- (9) Di Liberto, G.; Cipriano, L. A.; Pacchioni, G. Universal Principles for the Rational Design of Single Atom Electrocatalysts? Handle with Care. *ACS Catal.* **2022**, 5846–5856. <https://doi.org/10.1021/acscatal.2c01011>.

2. The perfect linear relationship can be observed between y1 and y2 axis in Figure 5b (Figure S60 b, c), did this implied something?

We thank the reviewer for raising this point. In fact, such a perfect linear relationship is due to the intrinsic dependence of these two descriptors ($\Delta_r G[{}^*NO_3 \rightarrow {}^*NO_2]$ and $\Delta_r G[{}^*NO_2 \rightarrow {}^*+NO_2^-]$). In these two reaction coordinates, only one surface intermediate species (*NO_2) was involved in both reactions and the nitrate and nitrite ions are a bulk species with a constant energy state that is independent from the catalytic sites. Therefore, with different M-N-C catalysts, the energy of *NO_2 and thus each

Δ_rG were different, but the summation of the above two Δ_rG was equal to $\Delta_rG[\text{NO}_3^- \rightarrow \text{NO}_2^-]$, which is a catalysts independent and constant value.

In figure 5b, we showed $\Delta_rG[*+\text{NO}_3^- \rightarrow *+\text{NO}_2^-]$ and $\Delta_rG[*+\text{NO}_2^- \rightarrow *+\text{NO}_2]$ on Y1 and Y2 axes, in order to give a multi-perspective on the energy landscape, including different pathways of nitrate adsorption and nitrite evolution. This can give a comprehensive view of the physical triggers for nitrite and ammonia selectivity.

The following description was added to the caption of Figure S60b for clarification.

“It should be noted that $\Delta_rG[*+\text{NO}_3^- \rightarrow *+\text{NO}_2^-]$ and $\Delta_rG[*+\text{NO}_2^- \rightarrow *+\text{NO}_2]$ were dependent terms, given the fact that they shared same surface intermediate species $*+\text{NO}_2^-$ and all other species in these two reaction coordinates were bulk items”

3. The trend of the catalytic performance shown in Figure 3d differed from the LSV result in Figure S28.

We thank the reviewer for pointing this out. First, the LSVs were performed with RDE set-up under diffusion free conditions and all other electrochemical data in this work including Figure 3d were collected with static electrodes (porous hydrophilic carbon paper). For electrocatalytic NO_3RR , RDE results tend to give a qualitative trend of the performance of each catalyst at different cathodic potentials, while static electrodes can give a quantitative description on the selectivity and activity of each catalyst over a set electrolysis duration. Also, Figure S28 was an example of LSV for each catalyst, while Figure 3d was a collection of averaged selectivity's and activities with standard deviations. We believe this is the reason for the non-perfect match between LSV results and chronoamperometry measurements. The results in Figure 3d should represent the CA data given in Figures S30-S32 since this data is directly a function of the results in Figure 3d.

4. The structure of these M-N-C catalysts after the catalysis test should be checked.

We thank the reviewer for this point and while we acknowledge that XAS is largely preferred to characterize M-N-C catalysts, as we have done ex-situ on the pristine catalysts. Operando XAS studies on M-N-C catalysts under reductive potentials are very limited in the literature, mainly on Cu-N-C, Fe-N-C and Ni-N-C, however additional studies on Mn-, Co- and Pd-N-C were reported as discussed below.

First it is important to keep in mind that while all these reported catalysts are M-N-C, each is unique in its synthetic precursors and therefore final structure.

For Cu-N-C catalysts, it has been shown in several studies that under reductive potentials, the morphology of the Cu-N_x sites changes, forming Cu-nanoclusters.¹⁰⁻¹² The potential at which this significant restructuring happens is reported to be more cathodic than -0.6V vs. RHE. Additionally, in these Cu-N-C works, where Cu-N_x site restructuring is occurring, this is directly observed in the corresponding electrochemical performance (CA curves) of these reports, something that would be directly observed in both the NO_3RR & NO_2RR experiments.

As a complement to this, for other operando XAS reports on M-N-C catalysts under reductive potentials, where M = Mn, Fe, Co, Ni and Pd, no restructuring of the M-N_x site was observed and the M-N_x sites remain in a non-metallic oxidation state.¹³⁻¹⁷ For the metals listed above this was confirmed through operando XAS measurements, while postmortem STEM has been performed for Mn, Ru, Rh, Ce and other rare-earth elements, which observed exclusively atomically dispersed active sites (for $\text{CO}_2\text{RR}/\text{N}_2\text{RR}$).¹⁸⁻²³

We agree that direct operando XAS of all catalysts would be the best scenario, however, this endeavor work be a significant, independent work of its own. As a compromise, to provide some insight into possible morphological changes or changes expected based on the literature, we have addressed this in the text (citing several studies on operando XAS and post-mortem STEM of M-N-C catalyst under reductive potentials). Although Cu-N_x sites are known to undergo restructuring under more reductive potentials, other M-N_x sites in the literature (with operando XAS) have not exhibited this behavior. The text in the manuscript has been modified to make it clear that although atomically dispersed sites were maintained in the post-mortem STEM and changes in the electrochemical NO₃/NO₂RR response were not observed (which could indicate morphological changes), the possibility of morphological changes, especially at higher potentials is something that should be kept in mind and future studies would be required for operando evaluation of these M-N-C catalysts.

In addressing these changes, we have evaluated the representative samples used throughout the main text, Fe-, Rh- and La-N-C by postmortem STEM and have provided this in Figure S42-S44.

We have made significant changes to the discussion in lines 279 – 311 in regards to the discussion regarding the stability of M-N-C catalysts under reductive potentials. Additionally, we have worked to ensure that the language used in the discussion provides appropriate caution and provide clarity for the readers. The changed paragraph is provided in detail below (lines 279-311).

“It was recently observed for the NO₃RR and previously reported for the CO₂RR that under a cathodic potential, atomically dispersed Cu-N_x sites can reduce to form metallic clusters and even nanoparticles at more cathodic potentials.^{43–45} Interestingly, additional investigations employing operando XAS over other atomically dispersed transition metal sites (M = Mn, Fe, Co, Ni and Pd) revealed a stable atomically dispersed M-N_x site, even under highly reductive potentials.^{46–49} Further studies employing post-mortem STEM observed atomically dispersed sites after reductive potentials in the CO₂ and N₂ reduction reactions, over Ru-, Rh-, Ce-N-C and other rare earth metals.^{50–54} To investigate possible morphological changes to the atomically dispersed nature of the catalysts in this work after the NO₃RR electrolysis, the representative Fe-, Rh- and La-N-C catalysts were imaged. Post-mortem STEM images in Figure S42-S44 confirm atomically dispersed nature of the Fe-, Rh- and La-N_x sites after the series of NO₃RR electrolysis from -0.2 to -0.8 V for 2 hours each. It is important to note that post-mortem STEM does not reveal possible in-situ restructuring under electrolysis conditions. However, as reported in a recent study employing Cu-N-C for the NO₃RR, as morphological reconstructions were occurring as a result of the cathodic potential, a significant increase in the current is readily observed over time in the electrochemical response (chronoamperometry curves) as the Cu-N_x sites transformed into metallic Cu clusters/particles.⁵⁵ Similar changes in the electrochemical response would be expected in the current study during both the NO₃RR and NO₂RR if significant morphological reconstruction was occurring, however, this was not observed. This may suggest, however not confirm without operando XAS, that even for Cu-N-C (which has been shown to be more susceptible to morphological changes under reductive potentials than other M-N-C catalysts), significant morphological reconstruction of the Cu-N_x sites may not be occurring. Perhaps because of variations in the stability of the Cu-N_x sites, originating from the different M-N-C synthesis approaches and precursor selections. This behavior is observed in the literature for Cu-N-C, by comparing operando studies under reductive potentials, for which some studies report active site reconstruction at -0.2 V vs. RHE⁵⁵ and others report no changes up to -0.6 V vs. RHE⁴⁴. Additionally, at the mild cathodic potential in this work of -0.2 V, at which the correlations (discussed in the following section) are most accurate, significant morphological reconstruction may not be expected

and was not observed over time in the electrochemical NO_3RR and NO_2RR performances. Although no M-N_x restructuring was observed through post-mortem STEM (Fe-, Rh- and La-N-C) or suggested by the electrochemical response, this cannot be totally ruled out and will require future operando XAS studies for a robust evaluation.”

Figure S 42. Post-mortem AC-STEM of the Fe-N-C catalyst after NO_3RR electrolysis at -0.2, -0.4, -0.6 and -0.8 V vs. RHE for 2 hours at each potential. High magnification darkfield images show the atomically dispersed Fe sites. Lower magnification darkfield and brightfield images showing the absence of metallic nanoparticles.

Figure S 43. Post-mortem AC-STEM of the Rh-N-C catalyst after NO_3RR electrolysis at -0.2, -0.4, -0.6 and -0.8 V vs. RHE for 2 hours at each potential. High magnification darkfield images show the atomically dispersed Rh sites. Lower magnification darkfield and brightfield images show the absence of metallic nanoparticles.

Figure S 44. Post-mortem AC-STEM of the La-N-C catalyst after NO_3RR electrolysis at -0.2, -0.4, -0.6 and -0.8 V vs. RHE for 2 hours at each potential. High magnification darkfield images show the atomically dispersed La sites. Lower magnification darkfield image showing the absence of metallic nanoparticles.

- (10) Yang, J.; Qi, H.; Li, A.; Liu, X.; Yang, X.; Zhang, S.; Zhao, Q.; Jiang, Q.; Su, Y.; Zhang, L.; Li, J. F.; Tian, Z. Q.; Liu, W.; Wang, A.; Zhang, T. Potential-Driven Restructuring of Cu Single Atoms to Nanoparticles for Boosting the Electrochemical Reduction of Nitrate to Ammonia. *J. Am. Chem. Soc.* **2022**, *144* (27), 12062–12071. <https://doi.org/10.1021/jacs.2c02262>.
- (11) Weng, Z.; Wu, Y.; Wang, M.; Jiang, J.; Yang, K.; Huo, S.; Wang, X. F.; Ma, Q.; Brudvig, G. W.;

- Batista, V. S.; Liang, Y.; Feng, Z.; Wang, H. Active Sites of Copper-Complex Catalytic Materials for Electrochemical Carbon Dioxide Reduction. *Nat. Commun.* **2018**, *9* (1), 1–9. <https://doi.org/10.1038/s41467-018-02819-7>.
- (12) Karapinar, D.; Huan, N. T.; Ranjbar Sahraie, N.; Li, J.; Wakerley, D.; Touati, N.; Zanna, S.; Taverna, D.; Galvão Tizei, L. H.; Zitolo, A.; Jaouen, F.; Mougél, V.; Fontecave, M. Electroreduction of CO₂ on Single-Site Copper-Nitrogen-Doped Carbon Material: Selective Formation of Ethanol and Reversible Restructuration of the Metal Sites. *Angew. Chemie - Int. Ed.* **2019**, *58* (42), 15098–15103. <https://doi.org/10.1002/anie.201907994>.
- (13) Lu, C.; Jiang, K.; Tranca, D.; Wang, N.; Zhu, H.; Rodríguez-Hernández, F.; Chen, Z.; Yang, C.; Zhang, F.; Su, Y.; Ke, C.; Zhang, J.; Han, Y.; Zhuang, X. Electrochemical Reduction of Carbon Dioxide with Nearly 100% Carbon Monoxide Faradaic Efficiency from Vacancy-Stabilized Single-Atom Active Sites. *J. Mater. Chem. A* **2021**, *9* (44), 24955–24962. <https://doi.org/10.1039/d1ta05990d>.
- (14) Yang, H. Bin; Hung, S. F.; Liu, S.; Yuan, K.; Miao, S.; Zhang, L.; Huang, X.; Wang, H. Y.; Cai, W.; Chen, R.; Gao, J.; Yang, X.; Chen, W.; Huang, Y.; Chen, H. M.; Li, C. M.; Zhang, T.; Liu, B. Atomically Dispersed Ni(i) as the Active Site for Electrochemical CO₂ Reduction. *Nat. Energy* **2018**, *3* (2), 140–147. <https://doi.org/10.1038/s41560-017-0078-8>.
- (15) He, Q.; Lee, J. H.; Liu, D.; Liu, Y.; Lin, Z.; Xie, Z.; Hwang, S.; Kattel, S.; Song, L.; Chen, J. G. Accelerating CO₂ Electroreduction to CO Over Pd Single-Atom Catalyst. *Adv. Funct. Mater.* **2020**, *30* (17). <https://doi.org/10.1002/adfm.202000407>.
- (16) Leonard, N.; Ju, W.; Sinev, I.; Steinberg, J.; Luo, F.; Varela, A. S.; Roldan Cuenya, B.; Strasser, P. The Chemical Identity, State and Structure of Catalytically Active Centers during the Electrochemical CO₂ Reduction on Porous Fe-Nitrogen-Carbon (Fe-N-C) Materials. *Chem. Sci.* **2018**, *9* (22), 5064–5073. <https://doi.org/10.1039/c8sc00491a>.
- (17) Wang, M.; Liu, S.; Qian, T.; Liu, J.; Zhou, J.; Ji, H.; Xiong, J.; Zhong, J.; Yan, C. Over 56.55% Faradaic Efficiency of Ambient Ammonia Synthesis Enabled by Positively Shifting the Reaction Potential. *Nat. Commun.* **2019**, *10* (1), 1–8. <https://doi.org/10.1038/s41467-018-08120-x>.
- (18) Zou, H.; Rong, W.; Wei, S.; Ji, Y.; Duan, L. Regulating Kinetics and Thermodynamics of Electrochemical Nitrogen Reduction with Metal Single-Atom Catalysts in a Pressurized Electrolyser. *Proc. Natl. Acad. Sci. U. S. A.* **2020**, *117* (47), 29462–29468. <https://doi.org/10.1073/pnas.2015108117>.
- (19) Tao, H.; Choi, C.; Ding, L. X.; Jiang, Z.; Han, Z.; Jia, M.; Fan, Q.; Gao, Y.; Wang, H.; Robertson, A. W.; Hong, S.; Jung, Y.; Liu, S.; Sun, Z. Nitrogen Fixation by Ru Single-Atom Electrocatalytic Reduction. *Chem* **2019**, *5* (1), 204–214. <https://doi.org/10.1016/j.chempr.2018.10.007>.
- (20) Liu, J.; Kong, X.; Zheng, L.; Guo, X.; Liu, X.; Shui, J. Rare Earth Single-Atom Catalysts for Nitrogen and Carbon Dioxide Reduction. *ACS Nano* **2020**, *14* (1), 1093–1101. <https://doi.org/10.1021/acsnano.9b08835>.
- (21) Han, L.; Liu, X.; Chen, J.; Lin, R.; Liu, H.; Lü, F.; Bak, S.; Liang, Z.; Zhao, S.; Stavitski, E.; Luo, J.; Adzic, R. R.; Xin, H. L. Atomically Dispersed Molybdenum Catalysts for Efficient Ambient Nitrogen Fixation. *Angew. Chemie* **2019**, *131* (8), 2343–2347. <https://doi.org/10.1002/ange.201811728>.
- (22) Han, L.; Hou, M.; Ou, P.; Cheng, H.; Ren, Z.; Liang, Z.; Boscoboinik, J. A.; Hunt, A.; Waluyo, I.; Zhang, S.; Zhuo, L.; Song, J.; Liu, X.; Luo, J.; Xin, H. L. Local Modulation of Single-Atomic Mn Sites

for Enhanced Ambient Ammonia Electrosynthesis. *ACS Catal.* **2021**, *11* (2), 509–516. <https://doi.org/10.1021/acscatal.0c04102>.

- (23) Zhang, W.; Qin, X.; Wei, T.; Liu, Q.; Luo, J.; Liu, X. Single Atomic Cerium Sites Anchored on Nitrogen-Doped Hollow Carbon Spheres for Highly Selective Electroreduction of Nitric Oxide to Ammonia. *J. Colloid Interface Sci.* **2023**, *638*, 650–657. <https://doi.org/10.1016/j.jcis.2023.02.026>.

5. The title/unit in many figures was wrong, such as in Figure 4e, Figure S29, S32, and S54-S57.

Thank you to the reviewer for catching these errors. We have fixed issues with Figure 4e and issues in Figure S29-S32 and an analogous error was fixed on Figure S50-S53. We have addressed issues found in Figure S59. We really appreciate this comment as we would like to adhere to clarity in units.

Reviewer #2 (Remarks to the Author):

The authors provide a suitable response and revision to the reviewers' concerns. Thus, this manuscript can be accepted at its present version.

Reviewer #3 (Remarks to the Author):

The authors have thoroughly addressed all my concerns and comments, as well as those from the other reviewers, and the work is greatly improved as a result.

This work will contribute valuably to the field of NO₃RR and be of great interest to the readership of Nature Communications.

It is evident that Reviewers #2 and #3 are fully satisfied.

We hope now that this paper will be accepted for publication in Nature Communications.